# Differentially Private Synthetic Data via APIs 4: Tabular Data

Toan Tran [1]   Arturs Backurs [* 2]   Zinan Lin [* 2]   Victor Reis [* 2]   Li Xiong [* 1]   Sergey Yekhanin [* 2]

## Abstract

This paper investigates the problem of generating synthetic tabular data with differential privacy (DP) guarantees, enabling data sharing in sensitive domains. Despite extensive study, state-of-the-art methods often focus on minimizing low-order marginal query errors and overlook the challenges posed by high-order correlations. To address this gap, we extend the Private Evolution (PE) framework, originally developed for DP-compliant image and text synthesis, to tabular data. We introduce Tab-PE − an algorithm for synthetic tabular data generation under DP constraints. Tab-PE iteratively improves a candidate dataset via an evolutionary process that leverages tabular-specialized operators to produce variations, privately scores them, and selects the highest-quality samples to retain and propagate. In contrast to the original PE, which relies on large foundation models, Tab-PE employs heuristic operators with significantly lower computational costs, making PE more practical and scalable for tabular data. Through extensive experiments on real-world and simulation datasets, we demonstrate that Tab-PE substantially outperforms prior baselines on datasets exhibiting high-order correlations. Compared to the best baseline – AIM, Tab-PE improves classification accuracy by up to 10% while running $28\times$ faster.[1]

## 1. Introduction

Tabular data is a foundational data modality underlying applications across many domains. However, because it often contains sensitive information such as patient records and financial transactions, using and sharing such data are

challenging due to potential risks of exposing private information (Borisov et al., 2024). To tackle the privacy concerns, generating synthetic tabular data with differential privacy (DP) guarantees (Dwork, 2006) has been a long-standing and active research area (Li et al., 2014; Zhang et al., 2021; Liu et al., 2021; McKenna et al., 2022; Liu et al., 2023; Tran & Xiong, 2024; Cormode et al., 2025; Maddock et al., 2025; Rosenblatt et al., 2026). This synthetic data can be used for various purposes – such as data analysis, machine learning model training, and sharing with third parties – while still providing formal privacy guarantees for individual records in the original dataset.

Despite the potential, generating realistic tabular data remains challenging due to difficulties in capturing complex multi-dimensional data distributions under the privacy constraints. State-of-the-art (SOTA) methods (McKenna et al., 2022; Liu et al., 2021; 2023) address this by estimating low-order statistical queries (typically marginals) and then stitching them together to approximate the full data distribution. It is known that these marginal-based methods have a fundamental limitation: they do not scale well to model high-order correlations as the number of queries grows exponentially with the order (i.e., the number of relevant attributes) (Hu et al., 2023; Ponomareva et al., 2025). Since DP requires adding noise to each query answer, the noise accumulates as the number of queries increases. Therefore, estimating a large number of queries under strict privacy constraints is challenging and often leads to low-quality measurements. Despite the theoretical limitation, marginal-based methods still achieve strong empirical performance on existing benchmarks (Chen et al., 2025b; Tao et al., 2022).

We found that ***most prior evaluations sidestep the challenge of high-order correlations***. Datasets widely used in the literature appear to be dominated by low-order dependencies. Intuitively, we measure the order of correlations in a dataset by considering the downstream performance gap of simple classifiers that capture only low-order correlations (e.g., shallow decision trees) versus complex classifiers that leverage high-order correlations (e.g., deep trees). When the performance gap between these two types of classifiers is small, the dataset primarily reflects low-order correlations. Indeed, many commonly used datasets such as Adult (Becker & Kohavi, 1996), Bank (Moro et al., 2014), and Census (Ding et al., 2021) have this property. Varying the maximum depth

---

*Equal contribution; author order is alphabetical [1]Emory University [2]Microsoft Research.

*Proceedings of the $43^{rd}$ International Conference on Machine Learning*, Seoul, South Korea. PMLR 306, 2026. Copyright 2026 by the author(s).

[1]Our code is available at https://github.com/microsoft/DPSDA

of the XGBoost trees (Chen & Guestrin, 2016) yields trivial performance differences (typically <1%) (Fig. 9, App. C.1). This characteristic makes the existing leading methods using statistical queries appear highly effective, even though they do not model high-order correlations. Consequently, much of the field has been implicitly optimized for this favorable setting, while leaving open the question of whether the current methods can truly preserve high-order correlations that are not revealed by standard benchmarks.

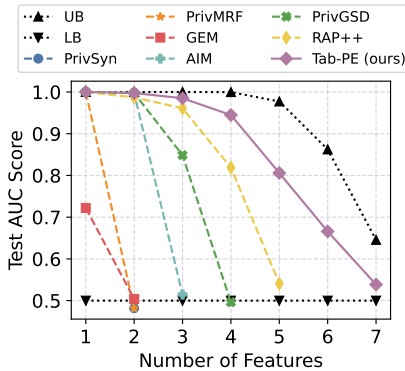

*Figure 1.* Stress test for high-order correlation modeling with XOR simulation datasets at $\epsilon = 1.0$. The binary label is assigned based on the parity of the number of positive features among all features, which requires capturing full-order correlations. UB stands for Upper Bound using private data. LB represents random guess.

In this work, we focus on investigating this gap. We construct a stress test with XOR correlations, where labels are assigned by the parity of the number of positive features across all dimensions, requiring to capture full-order correlations. We show that SOTA methods quickly fail to capture such high-order correlations (Fig. 1). To address this challenge, we propose a method based on the Private Evolution (PE) framework (Lin et al., 2024), tailored for tabular data – named Tab-PE. PE is a breakthrough that has shown promising results in generating high-quality synthetic data in other domains such as images (Lin et al., 2024; 2025; Wang et al., 2025a) and texts (Xie et al., 2024; Hou et al., 2024; 2025; Wang et al., 2025b). It generates synthetic data through an iterative process of generating variations of the data and then selecting the best ones based on a DP voting mechanism. Previous methods proposed different ways to generate variations of images or text such as using foundation models (Lin et al., 2024; Xie et al., 2024; Wang et al., 2025b) or using simulators (Lin et al., 2025). For tabular data, Swanberg et al. (2025) argue that Private Evolution using LLMs for tabular generation does not perform satisfactorily.

***Remark 1 (API).*** Following prior PE terminology, we use "API" to mean a callable generation interface, not necessarily an external service. In Tab-PE, APIs are lightweight local operators that do not rely on external foundation models or expensive services.

We design simple yet effective and efficient heuristic operators/APIs for generating variations of tabular data *without using any foundation models*. Building on the PE framework, Tab-PE first initializes a random synthetic dataset and then iteratively refines it. In each iteration, we generate variations by simply adding controlled random noise to numerical features and resampling categorical features with a scheduled probability. The synthetic samples are then scored by a DP voting mechanism based on full-record nearest-neighbor matching to private data, which can implicitly capture complex, high-dimensional dependencies. High-scoring samples are selected for the next iteration, enabling an iterative refinement process. We show that Tab-PE outperforms SOTA methods on a wide range of settings and is the most computationally efficient. Overall, our contributions can be summarized as follows:

- We revisit the challenge of modeling high-order correlations in differentially private synthetic tabular data generation. Our stress test reveals that SOTA methods fail to capture such correlations.

- We propose Tab-PE, a method based on the Private Evolution framework, with simple yet effective and efficient evolutionary operators for generating variations of tabular data without using any foundation models.

- We provide a broad collection of new datasets and settings that better reflect high-order correlations for evaluating differentially private tabular data generation methods, going beyond the standard benchmarks that are dominated by low-order correlations.

- Extensive experiments demonstrate that Tab-PE consistently outperforms the baselines for high-order fidelity and downstream utility, especially under strict privacy regimes. Tab-PE is also the most computationally efficient method and up to $28\times$ faster than utility-competitive baselines without requiring GPUs.

## 2. Related Works

**Differentially Private Tabular Synthesis**. DP synthetic tabular data is a long-standing problem with many prior works (Yang et al., 2024; Cormode et al., 2025), spanning the general setting, constrained generation (Ge et al., 2021), federated settings (Maddock et al., 2024), and relational databases (Alimohammadi et al., 2025; Pang et al., 2024). In a real-world competition (NIST, 2018), the winning solutions are dominated by methods that rely on marginal queries such as MST (McKenna et al., 2021), DPSyn (Li et al., 2021), and PrivBayes (Zhang et al., 2017). All these methods first answer the low-order marginal queries in a DP manner, then reconstruct the synthetic data from the noisy answers with different techniques, e.g., probabilistic graphical models (PGMs) (McKenna et al., 2019) and

Bayesian networks. To improve this pipeline, more advanced methods (AIM (McKenna et al., 2022), MRF (Cai et al., 2021)) dynamically select suitable marginal queries. Subsequently, RAP (Liu et al., 2021), RAP++ (Vietri et al., 2022), PrivGSD (Liu et al., 2023), and PrivPGD (Donhauser et al., 2024) consider generation as an optimization process that iteratively refines the synthetic dataset to minimize the error on the noisy answers. Meanwhile, JAM (Fuentes et al., 2024) aims to utilize publicly available data. Maddock et al. (2025) and Chen et al. (2025a) focus more on efficiency and scalability, with performance comparable to SOTA methods. Beyond the methods using statistical queries, there is a line of research that leverages machine learning methods for this problem. Inspired by the success of image generation, some works employ GANs (Xie et al., 2018; Yoon et al., 2019). Some recent works explore transformer-based architectures (Castellon et al., 2023; Sablayrolles et al., 2023), and large-language models (Tran & Xiong, 2024; Rosenblatt et al., 2026). Although these approaches narrow the gap to marginal-based methods compared with GANs, they still lag behind the marginal-based methods. A recent benchmark (Chen et al., 2025b) confirms that the marginal-based methods still dominate the field. In this work, we revisit the problem from the perspective of high-order correlations and propose a new efficient and effective framework that does not rely on statistical queries or model training.

**Private Evolution**. PE is a breakthrough for synthetic data generation with DP. PE was first introduced by Lin et al. (2024) for images. Unlike previous synthesizers, which require model training/fine-tuning on private data (Kurakin et al., 2024; Dockhorn et al., 2023), PE instead leverages inference API access to pretrained foundation models. By employing an evolutionary process that iteratively refines the synthetic data, PE achieves SOTA results while being computationally efficient. Xie et al. (2024) extended PE to text, demonstrating its effectiveness by significantly outperforming LLM DP fine-tuning baselines. Zou et al. (2025) enhanced the performance for text by utilizing multiple LLMs via a weighted fusion mechanism. Moreover, the PE framework has been adapted to federated learning settings to reduce communication costs while achieving better utility for language modeling (Hou et al., 2024; 2025). Additionally, Zhang et al. (2025) modified PE for few-shot generation, while González et al. (2025) studied theoretical convergence aspects of PE. For tabular data, Swanberg et al. (2025) applied PE with LLM-guided APIs. However, the authors argue that PE with LLM API access does not perform satisfactorily. While our work does not contradict their message, we demonstrate that PE using heuristic operators (without any foundation models) and appropriate designs can be both effective and computationally efficient.

While most prior PE-based works rely on foundation models, Sim-PE (Lin et al., 2025) was the first to show that the PE framework can leverage arbitrary data generators, including non-differentiable generator tools such as simulators. Examples include computer graphics renderers for image generation and robotics simulators for robotics applications. This highlights a key distinction between the PE framework and traditional training-based or model-based approaches for DP synthetic data: PE is not limited to machine learning models and can naturally incorporate both model-based and non-model-based generators within the same framework. In this work, we further extend this idea to tabular data generation. While our random and variation APIs (Section 4) share some high-level similarities with Sim-PE (Lin et al., 2025)—generating random values in the random API and adding perturbations in the variation API—our APIs are even simpler as they do not require any external tools such as simulators.

## 3. Preliminaries

**Differential Privacy**. $(\epsilon, \delta)$-differential privacy (DP) is a property of a randomized algorithm $\mathcal{M}$ that guarantees that the output of $\mathcal{M}$ does not change much whether we add or remove any particular entry in the input. More precisely, given any two neighboring datasets $\mathcal{D}, \mathcal{D}'$ (one can be obtained from the other by deleting a single entry) and any possible set of outputs $S$, it holds that $\Pr[\mathcal{M}(\mathcal{D}) \in S] \leq e^\epsilon \Pr[\mathcal{M}(\mathcal{D}') \in S] + \delta$ (Dwork & Roth, 2014).

**High-order correlation**. We define high-order correlation through a multivariate extension of mutual information. Let $X = \{X_1, X_2, ...X_k\}$ be a set of random variables representing $k$ attributes in a dataset. The *total correlation* (also called *multi-information*) $I(X_1, X_2, ..., X_k)$ (Watanabe, 1960) quantifies the total amount of information shared among all $k$ attributes.

We call that a $k$**-way correlation** exists among the $k$ attributes if the mutual information between the set of $k$ attributes is significantly greater than the maximum mutual information of any subset of $k - 1$ attributes. Formally, we say that there is a $k$-way correlation among attributes $X_1, X_2, ..., X_k$ if:

$$G_k = I(X_1, \ldots, X_k) \\ - \max_{i \in \{1, \ldots, k\}} I(\{X_1, \ldots, X_k\} \setminus \{X_i\}) > \Delta \quad (1)$$

This definition captures the idea that the joint behavior of all $k$ attributes contains unique information, which is not well explained by any subset of $k - 1$ attributes. We provide a proposition in App. A.1 to show that a non-trivial gap of two tree-based predictors with max depth of $k$ and $k - 1$ implies the existence of $k$-way correlation. This connection motivates us to constrain tree-based predictors to determine whether a dataset exhibits high-order correlations.

# 4. Methodology – Tab-PE

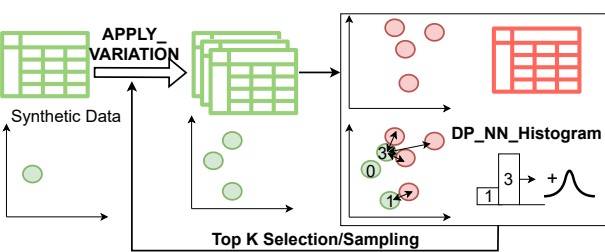

**Synthetic Data**

*Figure 2.* Illustration of Tab-PE's workflow. The process starts with an initial set of synthetic samples and iteratively refines them through variations and private scoring.

**Overview.** Sample $s = \{x_{\text{cat}(1)}, \dots, x_{\text{num}(1)}, \dots, c\}$ includes categorical attributes $x_{\text{cat}}$ and numerical attributes $x_{\text{num}}$, and class label $c$. $\mathcal{X}_{\text{cat}(i)}$ and $\mathcal{X}_{\text{num}(j)}$ denote the domains of categorical attribute $i$ and numerical attribute $j$, respectively. Given a private dataset $\mathcal{D}_{\text{priv}}$ of samples, our goal is to generate a synthetic dataset $\mathcal{D}_{\text{syn}}$ that preserves the statistical properties of $\mathcal{D}_{\text{priv}}$ while ensuring DP. Tab-PE consists of three main components: (1) a RANDOM_API that generates an initial set of synthetic samples, (2) a VARIATION_API that creates variations of existing samples to explore the sample space, and (3) a DP_NN_HISTOGRAM function that scores each sample in a DP manner. The overall process is an evolutionary loop, illustrated in Fig. 2. For each iteration, we generate variations of the current samples using VARIATION_API, evaluate them using the private dataset with DP_NN_HISTOGRAM, and retain the top-scoring samples to form the next generation. This process continues for a predefined number of iterations $T$.

**RANDOM_API**. The RANDOM_API generates an initial set of synthetic samples. For categorical attribute $x_{\text{cat}(i)}$, it samples a value uniformly at random from the set of possible categories $\mathcal{X}_{\text{cat}(i)}$. For numerical attribute $x_{\text{num}(j)}$, it uniformly samples values within the range $(\min_{\mathcal{X}_{\text{num}(j)}}, \max_{\mathcal{X}_{\text{num}(j)}})$. This ensures that the initial synthetic samples are diverse and cover the attribute space.

$$
\begin{aligned}
\texttt{RANDOM\_API}(n) &= \{s_1, s_2, \dots, s_n\} \text{ where} \\
s_k &= \{x_{\text{cat}(1)}, \dots, x_{\text{num}(1)}, \dots, c\}, \\
x_{\text{cat}(i)} &\sim \text{Uniform}(\mathcal{X}_{\text{cat}(i)}), \\
x_{\text{num}(j)} &\sim \text{Uniform}\big(\min(\mathcal{X}_{\text{num}(j)}), \max(\mathcal{X}_{\text{num}(j)})\big)
\end{aligned}
$$

**VARIATION_API**. The operator generates perturbed variations of existing samples to explore the sample space. The variation degree $m$ is the number of variations generated per sample. We implement a simple but effective *random walk* strategy. For a categorical attribute $x_{\text{cat}(i)} \in \mathcal{X}_{\text{cat}(i)}$, the variation is produced by resampling from its domain with a controlled categorical mutation rate $\mu_{\text{cat}} \in [0, 1]$.

$$
x'_{\text{cat}(i)} \sim \begin{cases} x_{\text{cat}(i)}, & \text{with probability } 1 - \mu_{\text{cat}}, \\ \text{Uniform}(\mathcal{X}_{\text{cat}(i)}), & \text{with probability } \mu_{\text{cat}}. \end{cases} \tag{2}
$$

For a numerical attribute $x_{\text{num}(j)} \in \mathcal{X}_{\text{num}(j)}$, the variation is generated by adding controlled Gaussian perturbation with scale controlled by a numerical mutation rate $\mu_{\text{num}} \in [0, 1]$ and projecting back into the valid range:

$$
\begin{aligned}
x'_{\text{num}(j)} &= \Pi_{\mathcal{X}_{\text{num}(j)}}(x_{\text{num}(j)} + \phi), \quad \phi \sim \mathcal{N}(0, \tau^2), \\
\tau &= \mu_{\text{num}} \cdot \big(\max(\mathcal{X}_{\text{num}(j)}) - \min(\mathcal{X}_{\text{num}(j)})\big).
\end{aligned}
$$

where $\Pi_{\mathcal{X}}(\cdot)$ denotes projection onto the feasible range of $\mathcal{X}_{\text{num}(j)}$. To simplify, the numerical bounds are assumed known, as in the default configuration of a widely used DP library (Holohan et al., 2019). These bounds can be estimated from the private data using DP methods if necessary (Dwork & Roth, 2014). We also ensure the baselines have access to the same numerical bounds for a fair comparison in our experiments.

Both mutation rates $\mu_{\text{cat}}$ and $\mu_{\text{num}}$ follow a polynomial decay as a function of the iteration index $t$ to balance exploration and exploitation. In the early stages, higher mutation rates encourage exploration of the sample space, while in later stages, lower rates focus on refining high-quality samples.

$$
\mu = \mu_{\text{init}} - (\mu_{\text{init}} - \mu_{\text{final}}) \cdot (t/T)^{\gamma} \tag{3}
$$

---

**Algorithm 1** DP_NN_HISTOGRAM

---

1: **Input:** $\mathcal{D}_{\text{priv}}$, Population $P$, Noise multiplier $\sigma$
2: **Output:** Noisy histogram $hist$
3: $hist \leftarrow [0, 0, \dots, 0]$
4: **for** each sample $s \in \mathcal{D}_{\text{priv}}$ **do**
5:    $i \leftarrow \arg\min_j \text{distance}(s, P[j])$
6:    $hist[i] \leftarrow hist[i] + 1$
7: **end for**
8: **for** each index $i$ in $hist$ **do**
9:    $hist[i] \leftarrow hist[i] + \mathcal{N}(0, \sigma^2)$
10: **end for**
11: **return** $hist$

---

**DP_NN_HISTOGRAM**. The DP_NN_HISTOGRAM scores synthetic samples in a DP manner. At each iteration $t$, Tab-PE maintains a population $P$, which is a set of candidate synthetic samples, mainly generated by VARIATION_API. We denote a histogram $hist$, where each bin $hist[i]$ corresponds to a sample $P[i]$ in $P$. The value $hist[i]$ represents the count of private samples in $\mathcal{D}_{\text{priv}}$ whose nearest neighbor in $P$ is $P[i]$. The pseudocode of DP_NN_HISTOGRAM is presented in Algo. 1. For each sample in the private dataset $\mathcal{D}_{\text{priv}}$, we find its nearest neighbor in $P$ and increment the

---

**Algorithm 2** Tabular Private Evolution

---

1: **Input:** The set of classes $C$, Private dataset $\mathcal{D}_{\text{priv}}$, Noise multiplier $\sigma$, Number of iterations $T$, Number of sampling iterations $T_{\text{sampling}}$, Variation degree $m$, Number of synthetic samples $N$
2: **Output:** Synthetic dataset $\mathcal{D}_{\text{syn}}$
3: $\mathcal{D}_{\text{syn}} \leftarrow \emptyset$
4: **for** each class $c \in C$ **do**
5:     $\mathcal{D}_{\text{priv}}^{(c)} \leftarrow$ subset of $\mathcal{D}_{\text{priv}}$ of class $c$
6:     $N^{(c)} \leftarrow N \cdot |\mathcal{D}_{\text{priv}}^{(c)}|/|\mathcal{D}_{\text{priv}}|$ {# syn samples of class c}
7:     $\mathcal{P}_0 \leftarrow \texttt{RANDOM\_API}(N^{(c)})$ {Initialize a population}
8:     **for** $t = 0$ **to** $T - 1$ **do**
9:        $hist_t \leftarrow \texttt{DP\_NN\_HISTOGRAM}(\mathcal{D}_{\text{priv}}^{(c)}, P_t, \sigma)$
10:        **if** $t \leq T_{\text{sampling}}$ **then**
11:           $hist_t[i] \leftarrow \max(0, hist_t[i])$
12:           $prob[i] \leftarrow hist_t[i]/\sum_j hist_t[j]$
13:           $\mathcal{D}_t \leftarrow$ sample $N^{(c)}$ samples from $P_t$ with replacement according to $prob$
14:        **else**
15:           $\mathcal{D}_t \leftarrow$ top $N^{(c)}$ samples of $P_t$ by $hist_t$
16:        **end if**
17:        **if** $t < T_{\text{sampling}}$ **then**
18:           $P_{t+1} \leftarrow \texttt{VARIATION\_API}(\mathcal{D}_t, 1)$
19:        **else**
20:           $P_{t+1} \leftarrow \texttt{VARIATION\_API}(\mathcal{D}_t, m) \cup \mathcal{D}_t$
21:        **end if**
22:     **end for**
23:     $\mathcal{D}_{\text{syn}} \leftarrow \mathcal{D}_{\text{syn}} \cup \mathcal{D}_{T-1}$
24: **end for**
25: **return** $\mathcal{D}_{\text{syn}}$

---

corresponding bin (Algo. 1, Lines 4-7). To ensure DP, we add Gaussian noise to each bin of the histogram (Algo. 1, Line 9). As each private sample can only affect one bin, the sensitivity of this histogram query is 1. By adding noise drawn from $\mathcal{N}(0, \sigma^2)$ to each bin, we achieve $(\epsilon, \delta)$-DP, where $\epsilon$ and $\delta$ are determined by the noise multiplier $\sigma$ and the number of iterations $T$. The privacy analysis can be reused from the Gaussian mechanism and the composition theorem, as done in the original private evolution paper (Lin et al., 2024) and detailed in App. B.1. The distance metric between samples is the mixed-type distance defined as follows, where $\lambda$ is a hyperparameter to balance the contributions of categorical and numerical attributes.

$$\text{distance}(s_a, s_b) = \lambda \sum_i \mathbb{1}\left(x_{\text{cat}(i)}^{(a)} \neq x_{\text{cat}(i)}^{(b)}\right)$$
$$+ \sum_j \left(\frac{x_{\text{num}(j)}^{(a)} - x_{\text{num}(j)}^{(b)}}{\max_{\mathcal{X}_{\text{num}(j)}} - \min_{\mathcal{X}_{\text{num}(j)}}}\right)^2 \quad (4)$$

**Tabular Private Evolution**. The overall process of Tab-PE is summarized in Algo. 2. We first initialize a synthetic dataset $\mathcal{D}_{\text{syn}}$ with $\texttt{RANDOM\_API}$. Then we iteratively refine the synthetic samples over $T$ iterations. In each iteration, we generate a population of sample candidates using $\texttt{VARIATION\_API}$, score them with $\texttt{DP\_NN\_HISTOGRAM}$, and select the top samples to form the next generation. To enhance exploration and exploitation, we employ a two-stage approach: sampling with replacement in the early iterations, followed by ranking and selecting the top samples in later iterations (See Algo. 2). In the first $T_{\text{sampling}}$ iterations, we sample new synthetic samples based on the noisy histogram-based probabilities. The variation degree $m$ is set to 1 (Algo. 2, Line 18) to maintain a small population size, which yields higher average histogram counts (Algo. 1, Lines 4-7) and thus reduces sensitivity to noise (Algo. 1, Line 9). This leads to more reliable sampling probabilities (Algo. 2, Line 12). In the second stage, we set $m$ to a higher value to encourage local refinement. The population $P$ now includes both the variations and the previous selected samples (Algo. 2, Line 20). We then select the top $N^{(c)}$ samples based on their noisy histogram scores (Algo. 2, Line 15). Intuitively, at the beginning, some samples may have significantly large counts and sampling with replacement allows these samples to be selected multiple times, which helps to quickly shift the distribution of synthetic samples towards the private data distribution. In the later stage, selecting the top samples helps to locally refine the synthetic dataset and improve its quality. This two-stage approach effectively exploits the strengths of both sampling and top selection, leading to better overall performance.

**Compute Efficiency**. The previous query-based methods require answering many queries. Each single query needs to scan the entire dataset. Moreover, high-dimensional queries involving many attributes are especially costly, as they create large multi-way count tables that consume significant memory and computation resources. Additionally, model fitting and optimization over these query measurements usually require iterative solvers that may scale poorly with the dimensionality. In contrast, Tab-PE operates at the sample level, and each iteration only requires a single pass over the private dataset to conduct nearest neighbor search. While the query-based methods struggle to handle high-order correlations due to the exponential growth of queries, Tab-PE leverages full-record nearest neighbor matching and iterative refinement that can implicitly capture complex, high-dimensional dependencies.

## 5. Experiments and Results

**Overview**. As we focus on high-order correlation modeling in DP tabular data, we first examine the algorithmic capa-

---

[1]We assume class distributions are known as in (Lin et al., 2024; Xie et al., 2024). Additionally, experiments in App. D.11 show Tab-PE performs similarly either w/ or w/o this assumption.

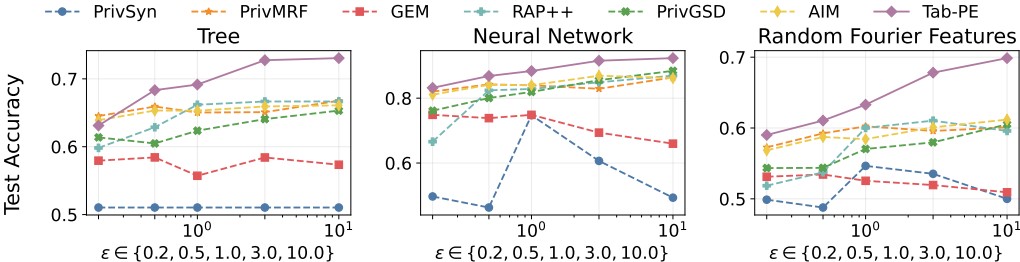

*Figure 3.* The test accuracy on SCM simulated datasets under various privacy budgets.

bility of the baselines and Tab-PE using an extreme case of XOR simulation datasets. We then conduct extensive experiments on realistic simulated datasets with multiple non-linear underlying functions and real-world datasets with high-order correlations, under various privacy constraint settings. We also evaluate the methods on widely-used real-world datasets with predominantly low-order correlations. Finally, we examine computational efficiency and analyze the technical design choices in Tab-PE.

### 5.1. Experiment Setup

**Baselines**. We consider several SOTA baselines in DP tabular data synthesizers, following a recent benchmark (Chen et al., 2025b): PrivSyn (Zhang et al., 2021), PrivMRF (Cai et al., 2021), GEM (Vietri et al., 2022), RAP++ (Liu et al., 2021), PrivGSD (Liu et al., 2023), and the SOTA method – AIM (McKenna et al., 2022). For details of these methods, we refer readers to the original papers and recent surveys (Yang et al., 2024; Cormode et al., 2025). Additionally, we present the upper bound performance (UB), directly using the private dataset without DP guarantees.

**Datasets**. In total, we organize the datasets used in our experiments into four categories. **1) XOR**, simulation stress-test datasets. **2) Structural Causal Model (SCM) Simulation** generated from causal graphs (details in App. C.1.2). **3) Real-World Datasets with High-Order Correlations**, in which complex classifiers *significantly* outperform simple ones, requiring synthetic data to capture high-order dependencies. **4) Real-World Datasets with Low-Order Correlations**, widely used in the literature, only low-order correlations are sufficient for high downstream accuracy.

**Evaluation Metrics**. Following previous benchmarks (Chen et al., 2025b; Tao et al., 2022; Du & Li, 2025), we evaluate the methods using Machine Learning (ML) Downstream Efficiency and Fidelity Error. For the fidelity, we calculate Wasserstein Distance (WD) between high-order joint distributions of the synthetic and private datasets, following prior work (Du & Li, 2025). Compared to another popular choice – Total Variation Distance (TVD), WD does not require discretizing continuous features into bins as TVD does. Moreover, WD is more reliable for high-order

joint distributions, while TVD is often distorted by sparse bins. Additionally, we perform evaluations in a unified embedding space, derived from an autoencoder trained on the private data with a reconstruction objective. This high-dimensional space enables us to compare the synthetic and real distributions at the representation level rather than just marginals. We calculate distributional Precision and Recall (Sajjadi et al., 2018), measuring the fidelity of synthetic samples to the real distribution and the coverage of the real distribution by the synthetic data, respectively. We present the details of the metrics in App. C.2

**Implementation Details**. We provide additional details and hyperparameters of Tab-PE in App. C.3. For the baselines, we follow the original papers and a recent benchmark (Chen et al., 2025b) for the hyperparameter settings. We run all methods on three distinct data splits generated by different random seeds and report the average performance values with corresponding standard deviations. When running an $(\epsilon, \delta)$-DP algorithm on a dataset $D_{priv}$ of size $|D_{priv}|$, for all methods, we set $\delta = 1/(|D_{priv}| \cdot \ln |D_{priv}|)$, which is a common choice in the DP literature (Yue et al., 2023).

### 5.2. Curse of Dimensionality

In this experiment, we examine the algorithmic capability of methods in modeling high-order correlation. We construct a simulated XOR dataset where all features are drawn from zero-centered uniform distributions. The label is assigned based on the parity of the number of positive feature values. In this dataset, the features themselves are mutually independent; the only dependency lies between the features and the label. This setup represents an extreme case where any single feature can flip the label. Consequently, failing to capture the contribution of only a single feature reduces the performance to random guessing (illustrated in Figs. 12 & 11, App. C.1.2). The baseline methods are set up with the ideal degree for marginal queries, i.e., $K = \text{num\_features} + 1$.

Fig. 1 presents the AUC score of the classifier trained on the synthetic data generated by the methods at $\epsilon = 1.0$. As the number of features increases, the classification problem itself becomes more challenging, leading to the performance drop of the upper bound – using private data ($\epsilon = \infty$). Intu-

| Dataset | Method | ML Downstream (↑) | | Fidelity (↓) | | | Embedding (↑) | |
|---|---|---|---|---|---|---|---|---|
| | | Accuracy | Macro F1 | 5-WD | 6-WD | 7-WD | Precision | Recall |
| Artificial Characters | *UB* | $80.80_{\pm 0.44}$ | $79.87_{\pm 0.65}$ | $0.088_{\pm 0.011}$ | $0.107_{\pm 0.011}$ | $0.124_{\pm 0.012}$ | $98.09_{\pm 0.15}$ | $98.66_{\pm 0.27}$ |
| | PrivSyn | $13.83_{\pm 0.00}$ | $2.43_{\pm 0.00}$ | $0.224_{\pm 0.005}$ | $0.287_{\pm 0.005}$ | $0.347_{\pm 0.006}$ | $13.42_{\pm 0.32}$ | $98.05_{\pm 0.43}$ |
| | PrivMRF | $13.63_{\pm 0.28}$ | $4.72_{\pm 3.24}$ | $0.218_{\pm 0.004}$ | $0.279_{\pm 0.004}$ | $0.337_{\pm 0.004}$ | $13.93_{\pm 0.18}$ | $97.33_{\pm 0.59}$ |
| | GEM | $10.13_{\pm 0.86}$ | $5.62_{\pm 0.46}$ | $0.337_{\pm 0.014}$ | $0.402_{\pm 0.015}$ | $0.465_{\pm 0.015}$ | $9.55_{\pm 0.75}$ | $94.57_{\pm 1.19}$ |
| | RAP++ | $33.29_{\pm 2.14}$ | $32.17_{\pm 2.11}$ | $0.201_{\pm 0.021}$ | $0.243_{\pm 0.023}$ | $0.283_{\pm 0.024}$ | $\underline{28.45}_{\pm 4.49}$ | $3.77_{\pm 1.76}$ |
| | PrivGSD | $40.36_{\pm 1.29}$ | $39.10_{\pm 1.38}$ | $\underline{0.161}_{\pm 0.002}$ | $\underline{0.204}_{\pm 0.002}$ | $\underline{0.245}_{\pm 0.002}$ | $26.98_{\pm 0.36}$ | $\mathbf{98.40}_{\pm 0.22}$ |
| | AIM | $23.24_{\pm 1.48}$ | $20.17_{\pm 1.24}$ | $0.191_{\pm 0.003}$ | $0.247_{\pm 0.002}$ | $0.301_{\pm 0.002}$ | $18.82_{\pm 0.55}$ | $\underline{98.06}_{\pm 0.21}$ |
| | Tab-PE | $\mathbf{49.38}_{\pm 0.46}$ | $\mathbf{48.09}_{\pm 0.71}$ | $\mathbf{0.158}_{\pm 0.011}$ | $\mathbf{0.191}_{\pm 0.012}$ | $\mathbf{0.220}_{\pm 0.013}$ | $\mathbf{36.57}_{\pm 1.51}$ | $89.77_{\pm 3.09}$ |
| Person Activity | *UB* | $78.01_{\pm 0.06}$ | $54.63_{\pm 0.36}$ | $0.108_{\pm 0.001}$ | $0.142_{\pm 0.001}$ | $0.176_{\pm 0.001}$ | $98.27_{\pm 0.13}$ | $98.30_{\pm 0.07}$ |
| | PrivSyn | $33.05_{\pm 0.00}$ | $4.52_{\pm 0.00}$ | $0.303_{\pm 0.003}$ | $0.385_{\pm 0.004}$ | $0.463_{\pm 0.004}$ | $41.87_{\pm 0.12}$ | $97.74_{\pm 0.12}$ |
| | PrivMRF | $51.83_{\pm 1.28}$ | $22.42_{\pm 1.01}$ | $0.138_{\pm 0.003}$ | $0.185_{\pm 0.004}$ | $0.233_{\pm 0.004}$ | $\underline{88.85}_{\pm 0.37}$ | $\underline{98.11}_{\pm 0.14}$ |
| | GEM | $31.85_{\pm 1.10}$ | $5.64_{\pm 0.79}$ | $0.275_{\pm 0.005}$ | $0.333_{\pm 0.006}$ | $0.392_{\pm 0.007}$ | $55.92_{\pm 3.42}$ | $95.20_{\pm 1.35}$ |
| | RAP++ | $52.72_{\pm 0.83}$ | $26.57_{\pm 0.82}$ | $0.176_{\pm 0.003}$ | $0.216_{\pm 0.003}$ | $0.256_{\pm 0.004}$ | $59.95_{\pm 2.49}$ | $62.36_{\pm 2.81}$ |
| | PrivGSD | $56.47_{\pm 0.36}$ | $29.25_{\pm 0.53}$ | $0.161_{\pm 0.001}$ | $0.199_{\pm 0.001}$ | $0.237_{\pm 0.002}$ | $80.06_{\pm 0.74}$ | $93.74_{\pm 0.58}$ |
| | AIM | $\underline{59.53}_{\pm 0.47}$ | $30.79_{\pm 0.32}$ | $\underline{0.125}_{\pm 0.002}$ | $\underline{0.168}_{\pm 0.002}$ | $\underline{0.213}_{\pm 0.002}$ | $\mathbf{89.97}_{\pm 0.24}$ | $\mathbf{98.73}_{\pm 0.07}$ |
| | Tab-PE | $\mathbf{63.72}_{\pm 0.18}$ | $\mathbf{35.09}_{\pm 0.19}$ | $\mathbf{0.116}_{\pm 0.002}$ | $\mathbf{0.150}_{\pm 0.002}$ | $\mathbf{0.183}_{\pm 0.001}$ | $90.93_{\pm 0.88}$ | $91.57_{\pm 0.38}$ |

*Table 1.* $\epsilon = 1.0$. The query degree hyperparameter of baselines vary from 2 to 5, the best-performing results of the baselines are reported.

itively, the number of marginal queries grows exponentially with the correlation order. For instance, assume each feature is binary-valued, to fully capture the correlation among $d$ features and the label, we need to answer $2^d$ queries of $d$-way marginals to cover all possible feature combinations. This is challenging to marginal query-based methods for modeling high-order correlations. Consequently, all the baselines fail completely at 5 features, delivering a downstream performance of random guess. In contrast, Tab-PE successfully yields an AUC score of 0.8 for 5 features. This demonstrates that **Tab-PE provides broader support for capturing high-order correlations**.

### 5.3. Simulated Datasets by Structural Causal Models

We adapt the simulation method from TabPFN (Hollmann et al., 2025), which is a breakthrough in tabular data classification. The full pipeline is described in App. C.1.2. Compared to the previous XOR setting, this simulation using Structural Causal Models (SCM) is a more realistic scenario: features are correlated; modeling a subset of the joint distribution can translate into gains for downstream tasks. We implement three non-linear prior functions (mapping features to labels): Tree, Neural Network (NN), and Random Fourier Features (RFF).

Across all prior functions, Tab-PE achieves the best downstream performance at $\epsilon = 1.0$. See Tab. 5, App. D.2 for numerical details. Tab-PE achieves 89.4% accuracy and 96.4% AUC for the neural network prior, significantly above the best baseline – AIM (85.2%, 93.3%). For fidelity (Tab. 6, App. D.2), AIM and Tab-PE offer competitive performance and outperform the other baselines. In the embedding space, Tab-PE consistently yields the highest precision ∼98% but the recall slightly lags at ∼81%. Overall, these results indicate that Tab-PE most effectively captures high-order

correlations to deliver the highest predictive downstream utility.

Fig. 3 depicts the test accuracy under different privacy-budget settings. In general, Tab-PE consistently outperforms the baselines under a variety of privacy settings. Most methods improve with larger $\epsilon$. Tree and RFF priors induce sharp, brittle high-order correlations that marginal-based methods cannot approximate well. This yields roughly 10% accuracy gains for Tab-PE over the best-performing baselines. In contrast, the NN prior often produces smoother correlations, so the gap remains around 4%. These results demonstrate that Tab-PE is effective at modeling challenging high-order correlations and maintains significant performance gains over baselines under either strict or loose privacy settings.

### 5.4. Real-world Datasets

We evaluate on two real-world datasets with high-order correlations (details in App. C.1). Generally, the performance trends are consistent with the previous SCM simulated datasets, as shown in Tab. 1. Tab-PE improves the downstream utilities by a large margin, e.g., +9.02% accuracy and +8.99% macro F1 on the Artificial Characters dataset, but still lags the non-private upper bound (∼30% accuracy gap). Moreover, consistent with the SCM datasets, Tab-PE achieves the highest precision in the embedding space. Tab-PE captures stronger high-order fidelity (from 5-way interactions onward), while being slightly behind AIM and MRF on low-order fidelity (1-way, 2-way, and 3-way marginals), detailed in App. D.3 (Tab. 7). The main reason is that marginal-based methods (e.g., AIM, MRF) explicitly optimize low-order marginals, making them excel in low-order fidelity. The fact that many high-order joint distributions can share the same 1-way and 2-way marginals further explains why preserving low-order fidelity well does

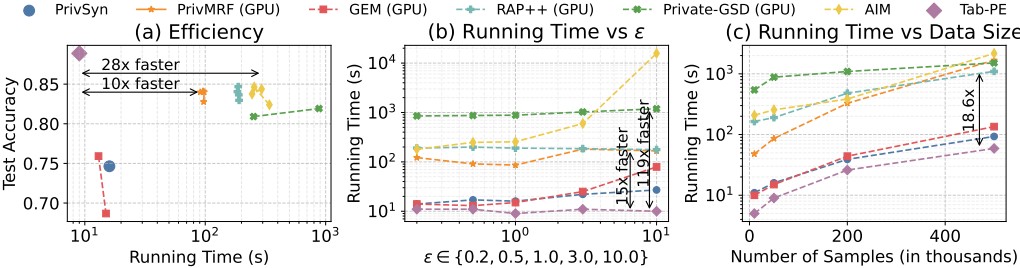

*Figure 4.* The runtime of the methods under different privacy budgets and dataset sizes. In the left figure, each method is shown with multiple markers, corresponding to various query degree settings. PrivSyn has only one marker as it does not have this hyperparameter.

not necessarily translate to high-order and downstream utilities. Conversely, better matching high-order patterns can be slightly inferior at low-order marginals, as the method does not focus on modeling low-order correlations. For example, consider a dataset with two binary features including 50% of "00" and 50% of "11". A method focusing on low-order marginals may generate samples with 50% of 01 and 50% of 10 and perfectly preserves 1-way marginals. Meanwhile, a method focusing on high-order correlations would generate 60% of 00 and 40% of 11, better capturing the joint distribution but slightly deviating from the 1-way marginals.

Moreover, Fig. 5 illustrates the test accuracy under different privacy budgets. Tab-PE consistently outperforms the baselines across the privacy settings. Due to space constraints, we present the results of low-order real-world datasets in App. D.4 (Tab. 8). While Tab-PE is primarily designed for high-order correlations, it remains competitive (only ∼1% accuracy drop compared to AIM) on low-order datasets.

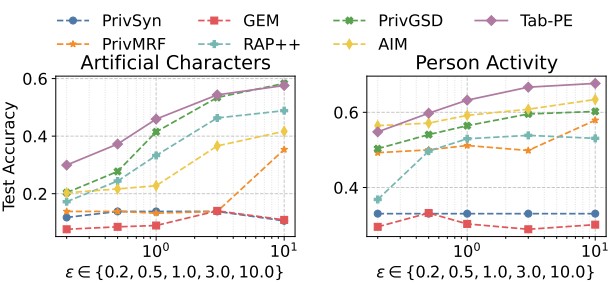

*Figure 5.* The test accuracy on real-world datasets under various privacy budgets.

### 5.5. Compute Efficiency & Scalability

We further study the compute efficiency and scalability of the methods. We run the experiments on the Neural Network prior simulation dataset using the same computing resources allocated by Slurm. While most baselines require GPUs, **Tab-PE runs entirely on CPUs**. As shown in Fig. 4 (left), at $\epsilon = 1.0$, Tab-PE is the most efficient method while achieving the best downstream utilities, $10\times$ faster than PrivMRF and $28\times$ faster than AIM. We also study the scalability of the baselines by varying the query

degree, detailed in App. D.5, Fig. 15. Generally, increasing the query degree does not bring significant performance gains for the baselines. However, it leads to an exponential increase in runtime for GEM and PrivGSD. Subsequently, most methods including Tab-PE scale well with the privacy budget, as presented in Fig. 4 (middle). Meanwhile, AIM exhibits a rapid increase ($60\times$) in runtime as $\epsilon$ increases from 1.0 to 10.0, as the larger privacy budget allows it to issue more queries. Finally, we examine the scalability of the methods with the dataset size. As depicted in Fig. 4 (right), at $\epsilon = 1.0$, Tab-PE is the fastest method across all dataset sizes. Notably, Tab-PE runs $18.6\times$ faster than the leading baselines (AIM, RAP++, GSD, and MRF) at 500K samples. Taken together, these results demonstrate that **Tab-PE is highly efficient and scalable**.

### 5.6. Findings & Analyses

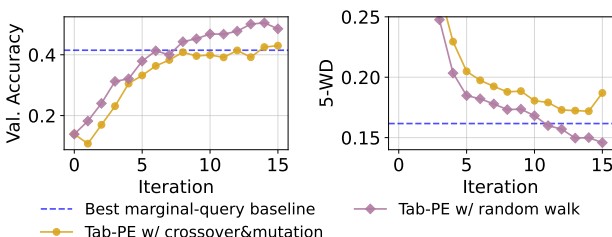

*Figure 6.* Comparing the proposed random-walk for variation generation with genetic algorithm operators.

**Simple variation operator can be effective.** We adapt the genetic algorithm design (crossover and mutation) from PrivGSD (Liu et al., 2023) to `VARIATION_API` in Tab-PE. The detailed implementations are provided in App. D.6.2. Fig. 6 shows that Tab-PE with either API achieves higher accuracy compared to the best marginal-based method (∼40%). The simple random walk with scheduled probability decay boosts accuracy by 7%, from 43% to 50%, and reduces 5-WD by around 21%.

**Two-stage selection outperforms ranking- or sampling-only strategies.** Fig. 7 presents the ablation study on two-stage selection by comparing with ranking-only and sampling-only strategies. While the sampling selection can

quickly preserve the distribution, which translates to the fidelity metric, the ranking selection is essential for local refinement to boost the downstream accuracy. The two-stage selection effectively combines the advantages of both strategies, leading to faster convergence and better performance.

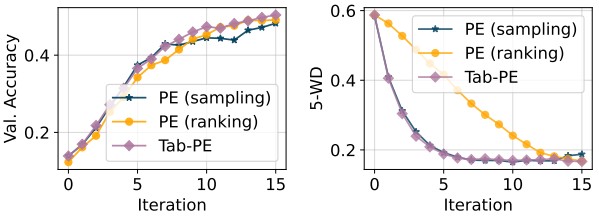

*Figure 7.* The performance of different selection strategies. Tab-PE implements a two-stage strategy: 5 iterations for sampling and 10 iterations for ranking (Artificial Characters; $\epsilon = 1.0$).

**Extremely High-Dimensional Dataset**. We experiment on flattened MNIST dataset with 196 attributes (rescaled to $14 \times 14$ pixels) and 10 classes. This dataset is not only high-dimensional but also expresses complex high-order correlations, since forming digit shapes requires a significant number of pixels to be jointly considered. It is worth noting that we do not consider image-based methods (such as CNNs) that leverage the spatial structure (i.e., pixel position) while tabular methods do not. The experiment details are described in App. D.7. At this scale, **most baselines are not computationally feasible** due to the large required GPU memory and runtime. Tab. 2 presents the test accuracy of the baselines and Tab-PE at $\epsilon = 1.0$. Tab-PE achieves a notable accuracy of 54%, while the query-based methods fail to capture the complex high-order correlations of pixels and deliver random guess performance.

| **Method** ($\epsilon = 1.0$) | GEM | RAP | Tab-PE |
|---|---|---|---|
| Test Accuracy (↑) | 12.06 | 9.82 | **54.05** |

*Table 2.* Classification accuracy on the flattened MNIST dataset.

**Hyperparameter Sensitivity Analysis**. We study the sensitivity of key hyperparameters in Tab-PE. The detailed results are presented in App. D.9. Generally, Tab-PE needs sufficient iterations (15-20) to converge and provide good utilities. Our ideal number of iterations is notably smaller than PrivGSD (also an evolutionary approach but operating on dataset level rather than sample level), which typically performs 200K iterations. The number of synthetic samples should be proportional to the dataset size to ensure an appropriate signal-to-noise ratio. At $\epsilon = 1.0$, Tab-PE performs best when generating 10-20% of the original size (Fig. 19), but the synthetic data can be further enriched by oversampling algorithms (App. D.10). The optimal hyperparameter setting is robust across $\epsilon$ settings (Fig. 25). We also show that Tab-PE is fairly robust to the choice of categorical-numerical weight $\lambda$. Additionally, we find

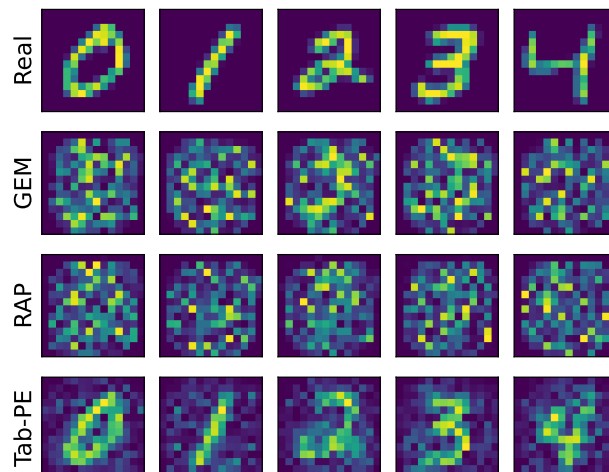

*Figure 8.* Samples generated by Tab-PE and the baselines with $\epsilon = 1.0$. Other digits are shown in Fig. 18, Sec. D.7.

that the polynomial decay schedule works better than the linear decay as it can allow more exploration in the early stage and more exploitation in the later stage, presented in App. D.6.1.

## 6. Conclusion

We revisit the challenges of modeling high-order correlations in synthetic tabular data generation under differential privacy constraints. We showed that existing methods struggle to capture these correlations. To address this, we introduced Tab-PE, a novel approach using Private Evolution. Our method effectively models high-order correlations while being lightweight and efficient. While Private Evolution has become an emerging paradigm showing promising performance for image and text synthesis (Lin et al., 2024; Xie et al., 2024; Lin et al., 2025), our exploration shows its potential in tabular data generation. With appropriate design choices, Tab-PE outperforms existing state-of-the-art methods in capturing high-order fidelity, downstream utility, and computational efficiency.

## Impact Statement

We believe our work has positive ethical implications. By enabling the generation of high-quality synthetic tabular data with differential privacy guarantees, our methods can enable data sharing, application, and innovation in many fields where privacy concerns currently limit data access. However, we also acknowledge the potential risks of synthetic data, even with DP guarantees, loose configurations of privacy parameters may still lead to information leakage. We encourage users to carefully consider the privacy-utility trade-offs and choose appropriate privacy parameters for their specific use cases.

## Acknowledgement

Toan Tran and Li Xiong are supported by the National Science Foundation under Award Numbers CNS-2437345, IIS-2302968, CNS-2124104, by the National Institutes of Health under Award Numbers R01ES033241 and R01LM013712. The views and opinions expressed in this paper are those of the authors and do not necessarily reflect the views of the U.S. Government or any agency thereof.

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

# Appendix of "Differentially Private Synthetic Data via APIs 4: Tabular Data"

Due to the space limit, we present additional details, results, and analyses in this Appendix.

## A. Preliminaries

### A.1. Analysis on high-order correlations

In this section, we show that constraining tree-based predictor depth can effectively quantify the order of correlations.

Let $Y$ be the label and $X = \{X_1, X_2, \ldots, X_{k-1}\}$ be the feature set. We denote $X_{-Z} = \{X_1, X_2, \ldots, X_{k-1}\} \backslash \{Z\}$ for $Z \in X$. Typically, the predictors are trained and evaluated by expected log-loss (i.e., cross entropy): $R_T = \mathbb{E}[-\log P_{f_T}(Y|X)]$, where $T$ is the depth of the tree-based predictor, and $f_T$ represents the predictor. We assume that each branch in the predictor $f_T$ can be an *arbitrary* function on *one* feature, and the predictor is trained to the optimal and $R_T^* = \min_{f_T} R_T$.

**Proposition A.1** (Performance gap of tree-based predictors indicates high-order correlations)**.** *Assume increasing the tree depth from $k-2$ to $k-1$ can significantly improve the prediction performance, i.e., the training loss decreases by $R_{k-2}^* - R_{k-1}^* > \Delta$, where $\Delta$ is a non-trivial positive constant.*

*Then the set of attributes $\{Y, X_1, \ldots, X_{k-1}\}$ exhibits a $k$-way correlation.*

**Proof.** Since $X$ contains only $k-1$ features, the predictor requires at most $k-1$ layers to achieve the optimal prediction. Therefore,

$$R_{k-1}^* = \min_{f_{k-1}} R_{k-1} = H(Y \mid X).$$

For the predictor $f_{k-2}$, each prediction branch can depend on at most $k-2$ features. Consequently, its prediction error is no worse than that of a predictor $f'_{k-2}$ that selects the best subset of $k-2$ features from $X$ so as to minimize the training loss. This implies

$$R_{k-2}^* \leq \min_{Z \in X} H(Y \mid X_{-Z}).$$

Combining the above inequalities, we obtain

$$\min_{Z \in X} H(Y \mid X_{-Z}) - H(Y \mid X) \geq R_{k-2}^* - R_{k-1}^* > \Delta.$$

Recall that, by definition, the set of attributes $\{Y, X_1, \ldots, X_{k-1}\}$ exhibits a $k$-way correlation if and only if

$$I(Y, X_1, \ldots, X_{k-1}) - \max_{Z \in \{Y, X_1, \ldots, X_{k-1}\}} I\big(\{Y, X_1, \ldots, X_{k-1}\} \setminus \{Z\}\big) > \Delta.$$

In what follows, we first show that

$$I(Y, X_1, \ldots, X_{k-1}) - \max_{Z \in \{X_1, \ldots, X_{k-1}\}} I\big(\{Y, X_1, \ldots, X_{k-1}\} \setminus \{Z\}\big) > \Delta,$$

and then prove that

$$I(Y, X_1, \ldots, X_{k-1}) - I(X_1, \ldots, X_{k-1}) > \Delta.$$

Together, these two inequalities establish the desired conclusion.

**First term.** For

$$I(Y, X_1, \ldots, X_{k-1}) - \max_{Z \in \{X_1, \ldots, X_{k-1}\}} I\big(\{Y, X_1, \ldots, X_{k-1}\} \setminus \{Z\}\big),$$

we have

$$
I(Y, X_1, \ldots, X_{k-1}) - \max_{Z \in \{X_1, \ldots, X_{k-1}\}} I\big(\{Y, X_1, \ldots, X_{k-1}\} \setminus \{Z\}\big)
$$

$$
= \min_Z I\big(X_Z; \{Y\} \cup X_{-Z}\big)
$$

$$
= \min_Z \Big( I(X_Z; X_{-Z}) + I(Y; X_Z \mid X_{-Z}) \Big)
$$

$$
\geq \min_Z I(X_Z; X_{-Z}) + \min_Z I(Y; X_Z \mid X_{-Z})
$$

$$
> 0 + \Delta
$$

$$
= \Delta,
$$

where we utilize the fact that $\min_Z I(Y; X_Z \mid X_{-Z}) = \min_{Z \in X} H(Y|X_{-Z}) - H(Y|X) > \Delta$.

**Second term.** For

$$
I(Y, X_1, \ldots, X_{k-1}) - I(X_1, \ldots, X_{k-1}),
$$

we obtain

$$
I(Y, X_1, \ldots, X_{k-1}) - I(X_1, \ldots, X_{k-1})
$$

$$
= I(Y; X_1, \ldots, X_{k-1})
$$

$$
= I(Y; X_{-X_1}) + I(Y; X_1 \mid X_{-X_1})
$$

$$
> 0 + \Delta
$$

$$
= \Delta,
$$

where we utilize the fact that $I(Y; X_1 \mid X_{-X_1}) \geq \min_Z I(Y; X_Z \mid X_{-Z}) > \Delta$.

This concludes the proof.

## B. Methodology

### B.1. Privacy Analysis

The privacy analysis of Algorithm 2 can be reused from (Lin et al., 2024) (see Section 4.3 there), as Tab-PE changes only non-private steps of the original PE framework. For completeness, we include it here as well. The DP guarantee of the Tab-PE algorithm (Algorithm 2) can be reasoned as follows:

- Step 1: The sensitivity of DP Nearest Neighbors Histogram (Algorithm 1). Each private sample only contributes one vote for histogram of one class. If we add or remove one sample, the resulting histogram for the corresponding class will change by at most 1 in the $\ell_2$ norm. Therefore, the sensitivity is upper bounded by 1.

- Step 2: Regarding each PE iteration as a Gaussian mechanism. The second for loop of Algorithm 1 adds i.i.d. Gaussian noise with standard deviation $\sigma$ to each bin. This is a standard Gaussian mechanism ((Dwork & Roth, 2014)) with noise multiplier $\sigma$.

- Step 3: Regarding the entire PE algorithm as $T$ adaptive compositions of Gaussian mechanisms, as Tab-PE is simply applying Algorithm 1 $T$ times sequentially.

- Step 4: Regarding the entire Tab-PE algorithm as one Gaussian mechanism with noise multiplier $\sigma/\sqrt{T}$. It is a standard result from (Dong et al., 2022) (see Corollary 3.3 therein).

- Step 5: Computing DP parameters $\epsilon$ and $\delta$. Since the problem is simply computing $\epsilon$ and $\delta$ for a standard Gaussian mechanism, we use the formula from (Balle & Wang, 2018) directly.

# C. Experimental Setup

## C.1. Datasets

### C.1.1. REAL-WORLD DATASETS

**Quantifying high-order correlation through classifier performance gap**   We aim to study how well the methods can capture high-order correlations. It is easy to be misled about high-dimensional correlations and high-dimensional datasets. While some datasets can have a large number of features, the features are often independent or only have low-order correlations (i.e., dependencies involving only a few features). We first propose a way to quantify the order of correlation in a dataset by considering the performance gap between simple classifiers, which only capture low-order correlations, and complex classifiers, which can leverage high-order correlations. The larger the gap, the more high-order correlations exist in the dataset. In practice, motivated by Proposition A.1, we vary the max depth of XGBoost, where the decision depth works as an upper bound on the order of captured correlations.

**Widely used datasets are dominated by low-order correlations**   We investigate a variety of datasets that have been widely used in prior evaluations (Chen et al., 2025b; Tao et al., 2022). We increase the max depth from 2 to 7, while keeping other hyperparameters as default. The results are shown in Figure 9. The gap of accuracy is trivial (typically smaller than 1%). This indicates that the downstream tasks on these datasets are dominated by low-order correlations. This leads to the conclusion that these datasets are not suitable for evaluating the ability to capture high-order correlations because synthesizers that can only capture low-order correlations may already achieve good performance.

**Datasets with high-order correlations**   We selected two datasets that yield significant differences in accuracy while varying the max depth of XGBoost, as depicted in Figure 10. In particular, we consider the Artificial Characters dataset (Guvenir et al., 1992) [2] and the Person Activity dataset (Vidulin et al., 2010) [3]. The Artificial Characters dataset contains 10218 samples with 8 numerical features and 10 classes, while the Person Activity dataset includes 164860 samples with 2 categorical features, 6 numerical features, and 11 classes.

### C.1.2. SIMULATION DATASETS

**XOR correlations as a stress test**   We consider XOR correlations as a stress test for capturing high-order correlations. The XOR function is a classic example that requires all input features to determine the output. Failing to model any single feature leads to random guessing. Each feature is drawn from a uniform distribution over $(-10, 10)$. The label is then determined by the parity of positive features.

$$c = \begin{cases} 1 & \text{if } \sum_{i=1}^{d} \mathbb{1}(x_i > 0) \text{ is odd} \\ 0 & \text{otherwise} \end{cases}$$

For each setting of the number of features, we generate 50K samples and ensure balanced binary classes. The dataset with two features is visualized in Figure 11. Figure 12 presents the performance of XGBoost classifiers with varying max depths on the XOR datasets. The max depth of XGBoost must be equal to the number of features to achieve better-than-random accuracy. Therefore, the synthetic data must capture the full high-order correlations to achieve good downstream utilities.

**SCM simulation datasets offer sustainable high-order correlations**   We adapt the simulation method from TabPFN (Hollmann et al., 2025), which is a breakthrough in tabular data classification. TabPFN generates large-scale realistic simulation data and pretrains a foundation model for in-context learning. By learning on only the simulation data, TabPFN still offers strong generalization to real-world data. This simulation pipeline employs Structural Causal Models (SCMs). An SCM defines a directed acyclic graph where each node corresponds to a feature, and the edges capture causal dependencies. The features are then generated by sampling values according to these dependencies that can represent complex interactions and non-linear relationships. The label is calculated by a prior function of features, inducing high-order correlations between the label and the feature set. As a result, increasing the max depth of XGBoost can lead up to a 10% accuracy gap (Figure 13, Appendix C.1). Compared to the previous XOR setting, this is a more realistic scenario: features are correlated; modeling a subset of the joint distribution can translate into gains for downstream tasks. In our experiments, we implement three

---

[2] https://www.openml.org/search?type=data&id=1459
[3] https://www.openml.org/search?type=data&id=1483

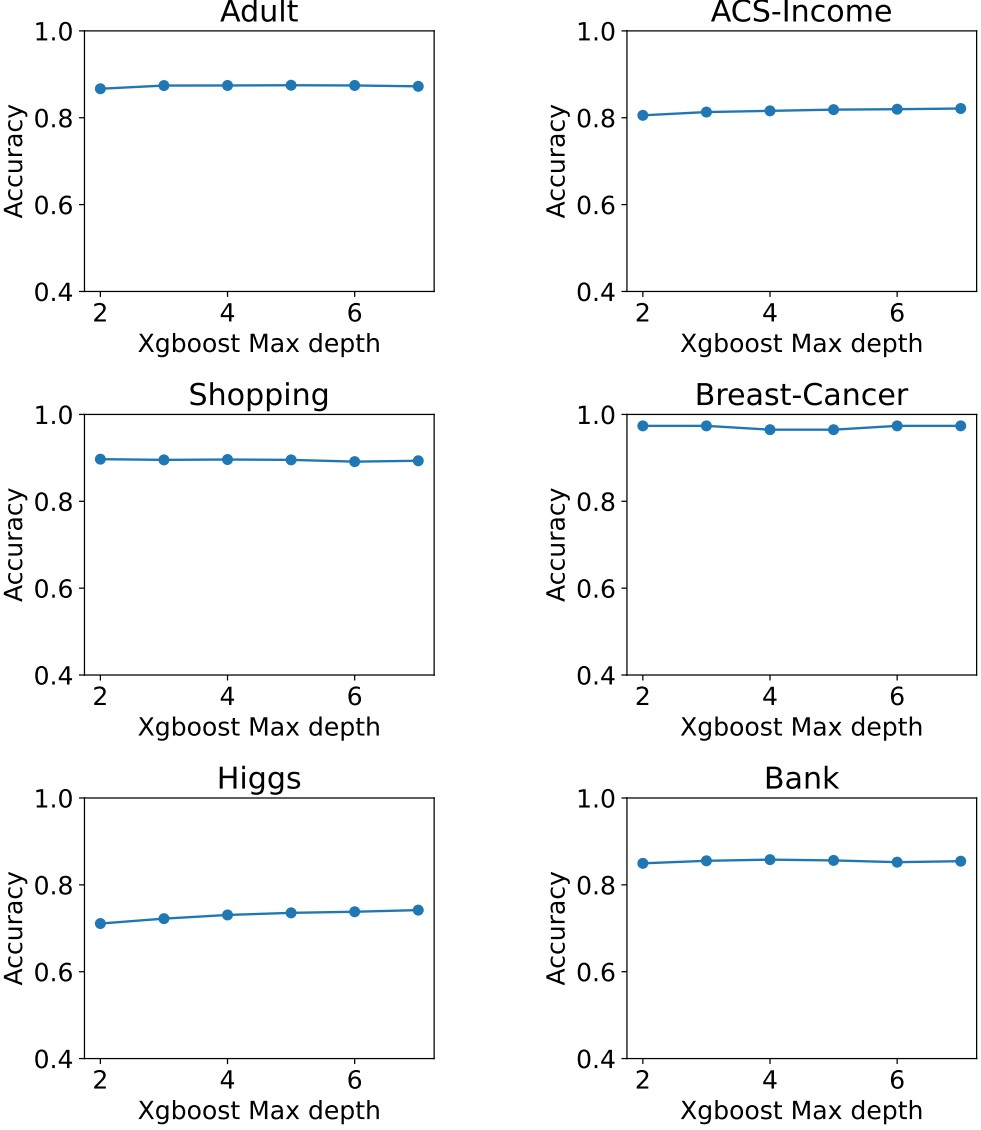

*Figure 9.* Datasets with low-order correlations. These are widely used in prior evaluations.

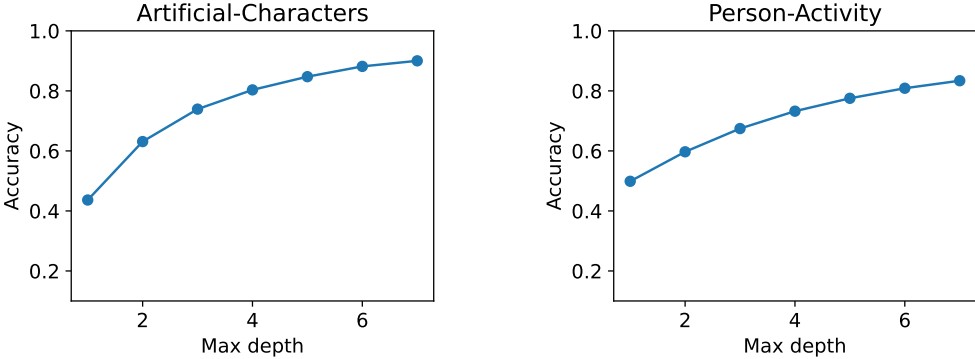

*Figure 10.* Datasets with high-order correlations – our focus.

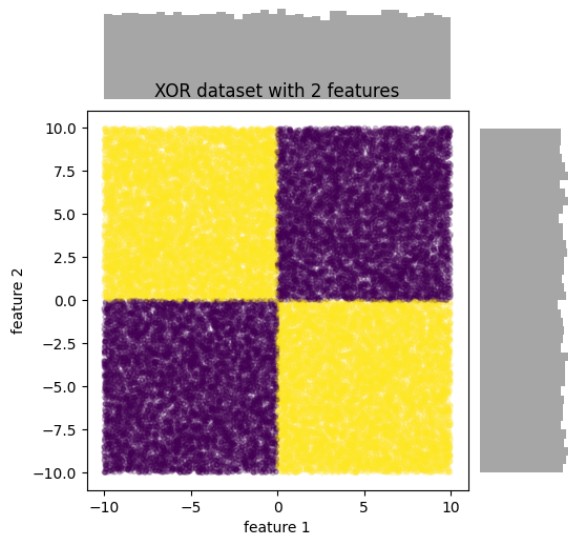

*Figure 11.* XOR dataset with 2 features. The colors represent classes.

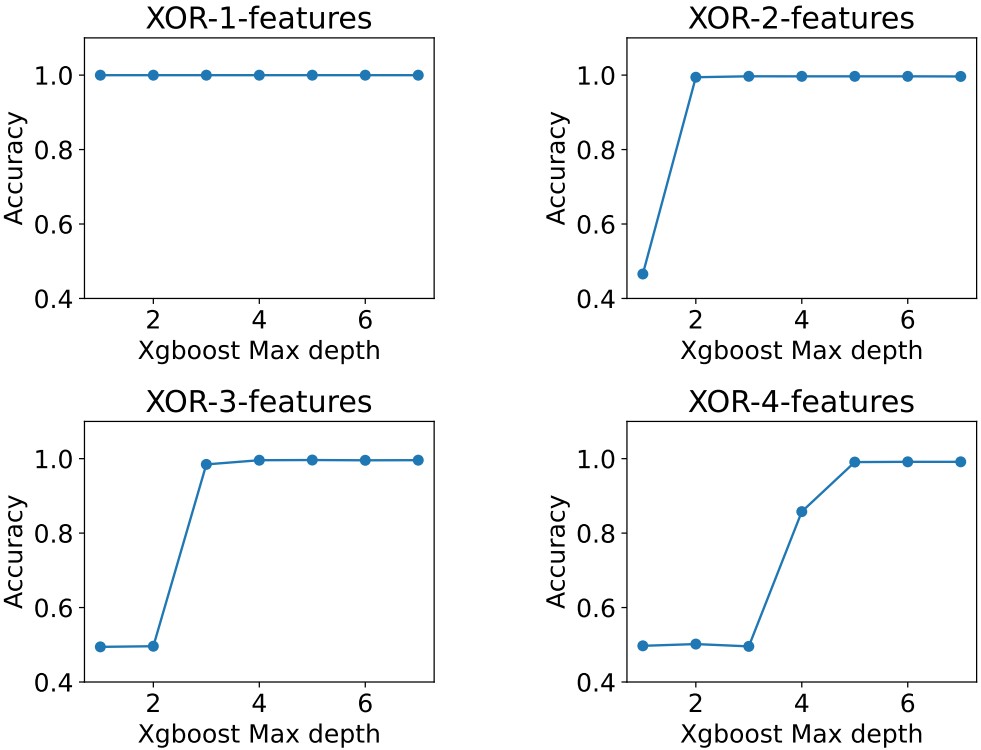

*Figure 12.* XOR Simulation Datasets.

non-linear prior function: Tree, Neural Network (NN), and Random Fourier Features (RFF). Each dataset includes 50K samples.

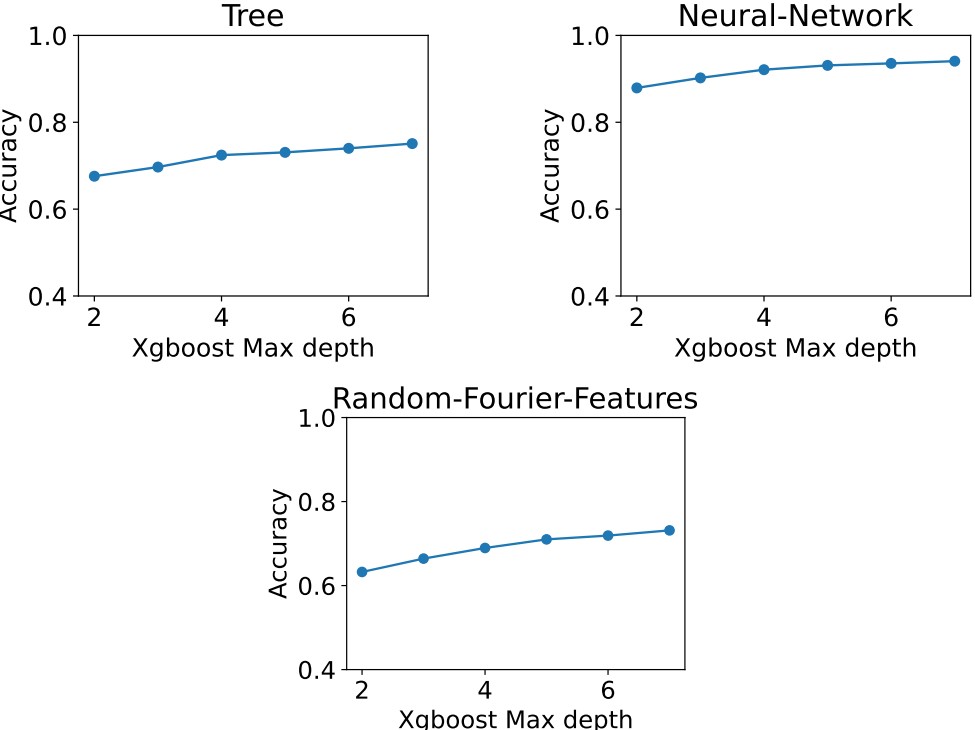

*Figure 13.* SCM Simulation Datasets.

## C.2. Evaluation Metrics

We consider the following metrics to evaluate the quality of synthetic data.

**Downstream utility**    The downstream utility reflects how well the synthetic data capture the correlation between features and labels. These metrics are the most important ones for studying high-order correlations. For consistency, we use the same SOTA classifier TabICL (Qu et al., 2025) for all datasets. TabICL is a transformer-based foundation model that has been pretrained on 82 million tabular datasets with in-context learning. It has demonstrated superior performance on a wide range of tabular datasets while **requiring little-to-no hyperparameter tuning**. For all methods, we fit TabICL on the generated synthetic data and evaluate on the *same* real test set.

**Fidelity of statistical properties**    To capture the statistical properties, we consider the Wasserstein distance. We denote $k$-WD as the average Wasserstein distance of all $k$-dimensional marginal distributions. The collection of $k$-attribute subsets:

$$\mathcal{C}_k \ = \ \{\, S \subseteq \mathcal{F} \ : \ |S| = k \,\},$$

where $\mathcal{F}$ is the set of features/attributes including the label. The average Wasserstein distance of $k$-dimensional marginals is defined as:

$$k - \mathrm{WD}(\mathcal{D}, \mathcal{D}') = \frac{1}{|\mathcal{C}_k|} \sum_{S \in \mathcal{C}_k} WD\big(\mathcal{D}(X_S), \mathcal{D}'(X_S)\big),$$

where $\mathcal{D}$ and $\mathcal{D}'$ are the real and synthetic datasets, respectively; $X_S$ is the subset of features in $S$; and $WD(\cdot, \cdot)$ is the Wasserstein distance between two distributions. We use the Python package POT to compute the Wasserstein distance.

**Representation-level alignment**    Evaluating the alignment on the representation space is common for text and image generation (Lin et al., 2024; Xie et al., 2024). The alignment reflects how well the synthetic data cover the real data distribution

in the representation space which can capture somewhat high-dimensional dependencies. While the representation space is achieved directly from foundation models in text and image domains, tabular data is challenged by strong distribution-shift across datasets. Therefore, we train an autoencoder for each dataset using the private dataset directly with a reconstruction loss. It is worth noting that the autoencoder here is only used for evaluation, and is not part of the synthesis process. By training on the private data, we ensure that the representation space is reliable and meaningful. We then calculate precision and recall (Sajjadi et al., 2018) on the embeddings of real and synthetic data. Precision measures how many generated samples are actually close to the real data manifold, while Recall calculates how many real samples are covered by the generated data. The formulas of precision and recall are as follows:

$$\text{Precision} = \frac{1}{|\mathcal{D}_{\text{syn}}|} \sum_{x \in \mathcal{D}_{\text{syn}}} \mathbb{1}(\exists y \in \mathcal{D}_{\text{real}}, \|\phi(x) - \phi(y)\|_2 \leq r_k(\phi(y), \phi(\mathcal{D}_{\text{real}})))$$

where $\phi$ is the encoder of the autoencoder that maps the raw data to the representation space; $r_k(\phi(y), \phi(\mathcal{D}_{\text{real}}))$ is the distance from $\phi(y)$ to its $k$-th nearest neighbor in the set $\phi(\mathcal{D}_{\text{real}})$. We set $k = 5$ in our experiments. Recall is defined symmetrically by swapping $\mathcal{D}_{\text{syn}}$ and $\mathcal{D}_{\text{real}}$.

$$\text{Recall} = \frac{1}{|\mathcal{D}_{\text{real}}|} \sum_{y \in \mathcal{D}_{\text{real}}} \mathbb{1}(\exists x \in \mathcal{D}_{\text{syn}}, \|\phi(y) - \phi(x)\|_2 \leq r_k(\phi(x), \phi(\mathcal{D}_{\text{syn}})))$$

### C.3. Implementations

Our code, datasets, and instructions are available at https://anonymous.4open.science/r/tabpe-A11C. We split each dataset into 70% training, 15% validation, and 15% test sets, determined by fixed random seeds. All the methods are fitted on the same training set and evaluated on the same test set. The validation set is used for hyperparameter tuning for all methods. We generally do not account for the privacy budget for hyperparameter tuning. For the baselines, we reuse the code from a recent benchmark (Chen et al., 2025b) and follow their hyperparameter settings. For baselines that require discretization for numerical features, we employ PrivTree (Zhang et al., 2016), which yields better performance than uniform binning, according to (Chen et al., 2025b). For baselines using statistical queries, we use marginal queries, as they are the most commonly used in prior work and the most important for capturing high-order correlations. Rather than fixing the degree of marginal queries at 2, as is common in many previous setups, we treat it as a tunable hyperparameter (ranging from 2 to 5), since our datasets exhibit high-order correlations. This tuning maximizes the chance of capturing such correlations. The other hyperparameters of the baselines are presented in Table 3.

By default, we run Tab-PE with the hyperparameters presented in Table 4 if not specified. For the real-world datasets, we generate 1K samples for the Artificial Characters dataset and 5K samples for the Person Activity dataset. For the simulation data, we generate 2K samples by default.

## D. Additional Results

### D.1. Data distribution of Tab-PE over iterations

Figure 14 illustrates the evolutionary process of synthetic datasets generated by Tab-PE. At the beginning (iteration 0), the synthetic data is mostly random. As the algorithm progresses, the synthetic data gradually aligns with the private data distribution.

### D.2. SCM Simulation Datasets

Table 5 presents the performance of all methods on the SCM simulation datasets. Tab-PE consistently outperforms all baselines for the downstream utility metrics. Table 6 presents the fidelity metrics. For the fidelity metrics, Tab-PE offers the best performance for the NN and RFF datasets, while AIM achieves the best for the Tree dataset. In the embedding space, Tab-PE achieves the best precision, demonstrating that the synthetic samples generated by Tab-PE are indeed close to the real data at the representation level. Overall, these results indicate the effectiveness of Tab-PE in capturing high-order correlations.

| Method | Hyperparameter | Value |
|---|---|---|
| PrivSyn | Consistent Iteration | 501 |
| | Max update iteration | 50 |
| PrivMRF | Graph construction parameter | 6 |
| | Sample size | 400 |
| | Estimation iteration | 3000 |
| | Size penalty | 1e-8 |
| | Max clique size | 1e+7 |
| GEM | Synthesis size | 1024 |
| | Learning rate | 1e-3 |
| | Max iteration | 500 |
| | Max selection round | $5 \cdot$ number of attributes |
| RAP++ | Random Projection Number | 2e+6 |
| | Categorical optimization rate | 3e-3 |
| | Numerical optimization rate | 6e-3 |
| | Top q | 5 |
| | Categorical optimization step | 1 |
| | Numerical optimization step | 3 |
| | Upsample rate | 10 |
| PrivGSD | Mutation rate | 50 |
| | Crossover rate | 50 |
| | Upsample number | 1e+5 |
| | Number of iterations | 1e+6 |
| AIM | Max model size | 100 |
| | Max iteration | 1000 |
| | Max marginal size | 2.5e+5 |

*Table 3.* Hyperparameters of the baselines.

| Hyperparameter | Value |
|---|---|
| Number of iterations $T$ | 15 |
| Number of sampling iterations $T_{\text{sampling}}$ | 5 |
| Variation degree $m$ | 3 |
| Mutation rate initial value $\mu_{\text{init}}$ | 0.5 |
| Mutation rate final value $\mu_{\text{final}}$ | 0.02 |
| Categorical mutation rate $\mu_{\text{cat}}$ | Polynomial decay from $\mu_{\text{init}}$ to 0.02 |
| Numerical mutation rate $\mu_{\text{num}}$ | Polynomial decay from $\mu_{\text{init}}$ to 0.02 |
| Decay factor $\gamma$ | 0.2 |
| Categorical weight $\lambda$ | 1/3 |
| Privacy budget $\epsilon$ | 1.0 |
| Privacy delta $\delta$ | $1/(|\mathcal{D}_{\text{real}}| \cdot \ln(|\mathcal{D}_{\text{real}}|))$ |

*Table 4.* Default hyperparameters of Tab-PE.

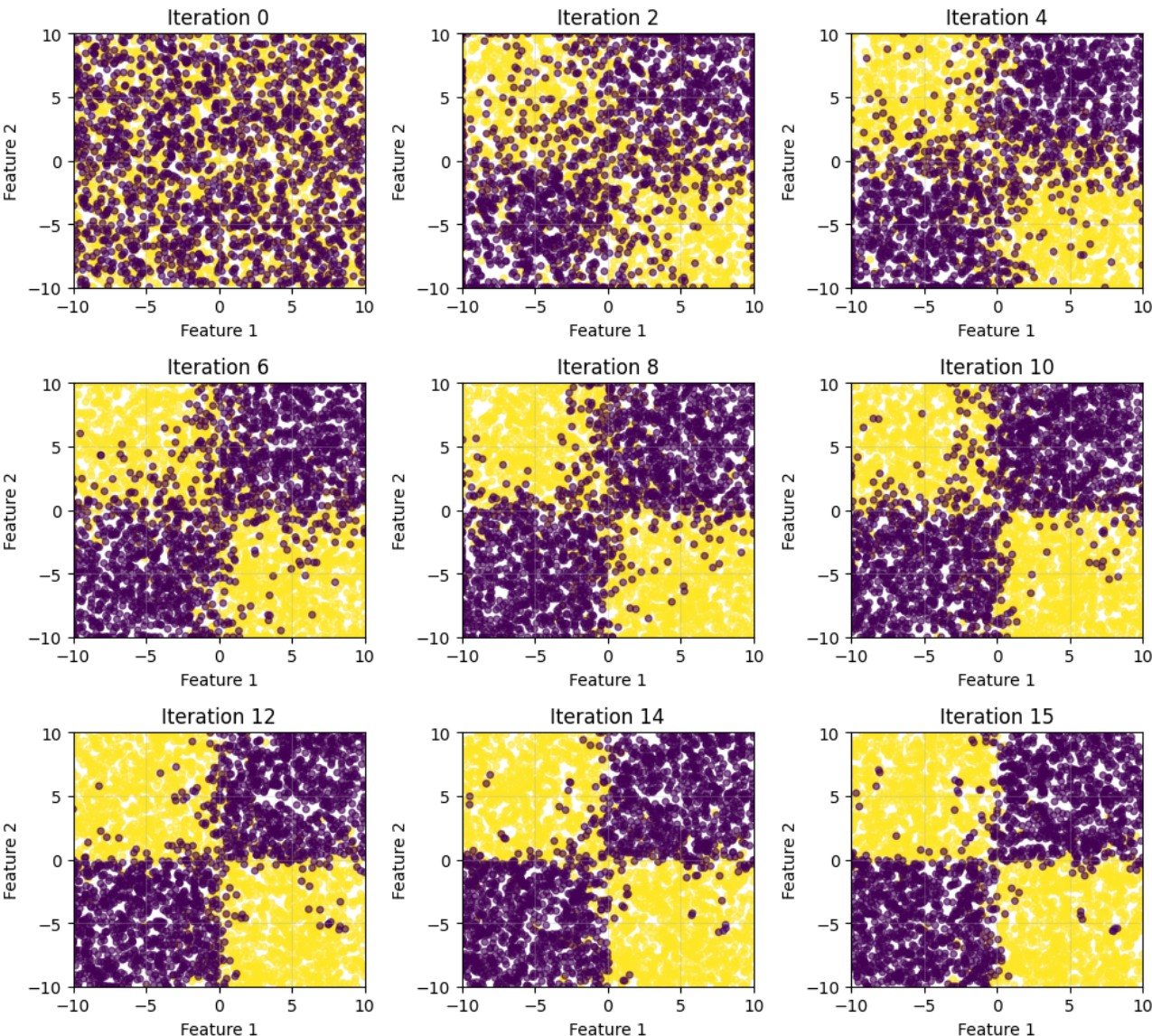

*Figure 14.* Synthetic Datasets generated by Tab-PE over iterations for the XOR dataset with 2 features.

| Prior | Method | ML Downstream (↑) | | Fidelity (↓) | | | Embedding (↑) | |
|---|---|---|---|---|---|---|---|---|
| | | Accuracy | AUC Score | 5-WD | 6-WD | 7-WD | Precision | Recall |
| Tree | *UB* | *81.41* ± *0.09* | *90.17* ± *0.45* | *0.126* ± *0.004* | *0.164* ± *0.005* | *0.202* ± *0.005* | *98.05* ± *0.22* | *97.74* ± *0.09* |
| | PrivSyn | 51.04 ± 0.00 | - | 0.182 ± 0.008 | 0.235 ± 0.007 | 0.288 ± 0.007 | 65.61 ± 0.98 | 98.08 ± 0.17 |
| | PrivMRF | 65.68 ± 0.52 | 71.26 ± 0.82 | 0.139 ± 0.001 | 0.178 ± 0.001 | 0.216 ± 0.000 | 92.11 ± 0.61 | **98.34** ± **0.13** |
| | GEM | 56.66 ± 0.67 | 60.47 ± 0.47 | 0.187 ± 0.008 | 0.241 ± 0.008 | 0.295 ± 0.008 | 59.65 ± 1.15 | 97.92 ± 0.18 |
| | RAP++ | 64.77 ± 1.09 | 69.77 ± 1.49 | 0.190 ± 0.013 | 0.234 ± 0.015 | 0.278 ± 0.015 | 93.21 ± 1.11 | 23.94 ± 3.32 |
| | PrivGSD | 61.99 ± 0.37 | 66.22 ± 0.85 | 0.141 ± 0.011 | 0.181 ± 0.011 | 0.221 ± 0.011 | 84.79 ± 0.77 | 91.74 ± 0.28 |
| | AIM | 65.40 ± 0.26 | 71.19 ± 0.10 | **0.128** ± **0.009** | **0.167** ± **0.009** | **0.206** ± **0.009** | 93.47 ± 0.45 | 98.33 ± 0.04 |
| | Tab-PE | **68.78** ± **0.30** | **75.24** ± **0.36** | 0.132 ± 0.008 | 0.170 ± 0.008 | 0.208 ± 0.009 | **98.63** ± **0.31** | 79.61 ± 1.33 |
| NN | *UB* | *96.58* ± *0.10* | *96.58* ± *0.10* | *0.119* ± *0.012* | *0.156* ± *0.012* | *0.192* ± *0.012* | *98.02* ± *0.16* | *97.77* ± *0.05* |
| | PrivSyn | 51.97 ± 16.18 | 50.59 ± 22.57 | 0.204 ± 0.010 | 0.255 ± 0.009 | 0.306 ± 0.008 | 66.31 ± 0.08 | 97.93 ± 0.36 |
| | PrivMRF | 84.78 ± 0.71 | 92.75 ± 0.77 | 0.131 ± 0.011 | 0.171 ± 0.013 | 0.211 ± 0.014 | 92.89 ± 0.29 | **98.41** ± **0.25** |
| | GEM | 74.26 ± 0.81 | 82.61 ± 0.47 | 0.219 ± 0.026 | 0.275 ± 0.024 | 0.330 ± 0.023 | 57.55 ± 0.76 | 98.24 ± 0.13 |
| | RAP++ | 85.16 ± 1.21 | 93.06 ± 1.10 | 0.179 ± 0.018 | 0.221 ± 0.020 | 0.263 ± 0.022 | 94.00 ± 0.26 | 21.81 ± 2.22 |
| | PrivGSD | 82.47 ± 0.40 | 90.86 ± 0.49 | 0.147 ± 0.004 | 0.187 ± 0.004 | 0.227 ± 0.005 | 84.96 ± 0.42 | 91.17 ± 0.56 |
| | AIM | 85.23 ± 0.40 | 93.26 ± 0.57 | 0.129 ± 0.014 | 0.168 ± 0.016 | 0.207 ± 0.017 | 93.10 ± 0.26 | 98.36 ± 0.23 |
| | Tab-PE | **89.36** ± **0.42** | **96.37** ± **0.25** | **0.125** ± **0.010** | **0.163** ± **0.010** | **0.199** ± **0.010** | **98.57** ± **0.39** | 81.27 ± 1.75 |
| RFF | *UB* | *81.12* ± *0.19* | *81.12* ± *0.19* | *0.113* ± *0.003* | *0.152* ± *0.003* | *0.191* ± *0.002* | *98.12* ± *0.18* | *97.55* ± *0.16* |
| | PrivSyn | 50.96 ± 2.61 | 50.56 ± 4.13 | 0.173 ± 0.005 | 0.227 ± 0.005 | 0.281 ± 0.005 | 66.83 ± 0.59 | 97.77 ± 0.44 |
| | PrivMRF | 60.11 ± 0.15 | 63.68 ± 0.29 | 0.122 ± 0.005 | 0.162 ± 0.005 | 0.202 ± 0.005 | 93.06 ± 0.16 | **98.55** ± **0.07** |
| | GEM | 53.76 ± 0.99 | 55.53 ± 0.95 | 0.187 ± 0.019 | 0.243 ± 0.019 | 0.298 ± 0.018 | 57.80 ± 3.26 | 98.51 ± 0.21 |
| | RAP++ | 59.00 ± 1.32 | 62.13 ± 1.72 | 0.178 ± 0.006 | 0.221 ± 0.005 | 0.263 ± 0.005 | 0.270 ± 0.007 | 25.84 ± 2.99 |
| | PrivGSD | 57.08 ± 0.07 | 59.70 ± 0.15 | 0.137 ± 0.004 | 0.179 ± 0.004 | 0.220 ± 0.003 | 85.13 ± 0.24 | 90.80 ± 0.79 |
| | AIM | 59.60 ± 1.32 | 62.76 ± 1.41 | 0.125 ± 0.004 | 0.165 ± 0.005 | 0.204 ± 0.005 | 93.41 ± 0.31 | 98.50 ± 0.16 |
| | Tab-PE | **64.10** ± **0.40** | **69.19** ± **0.45** | **0.120** ± **0.008** | **0.160** ± **0.008** | **0.198** ± **0.008** | **98.57** ± **0.39** | 81.27 ± 1.75 |

*Table 5.* The experiment is configured with $\epsilon = 1.0$. The degree hyperparameter of baselines varies from 2 to 5, the best-performing results of the baselines are reported. The best and second-best results are highlighted in **bold** and underline, respectively.

| Prior | Method | Fidelity (↓) | | | | | | |
|---|---|---|---|---|---|---|---|---|
| | | 1-WD | 2-WD | 3-WD | 4-WD | 5-WD | 6-WD | 7-WD |
| Tree | *UB* | *0.026* ± *0.002* | *0.046* ± *0.003* | *0.070* ± *0.004* | *0.089* ± *0.004* | *0.126* ± *0.004* | *0.164* ± *0.005* | *0.202* ± *0.005* |
| | PrivSyn | 0.030 ± 0.004 | 0.064 ± 0.008 | 0.103 ± 0.009 | 0.131 ± 0.008 | 0.182 ± 0.008 | 0.235 ± 0.007 | 0.288 ± 0.007 |
| | PrivMRF | 0.030 ± 0.003 | 0.051 ± 0.005 | 0.076 ± 0.006 | 0.102 ± 0.002 | 0.139 ± 0.001 | 0.178 ± 0.001 | 0.216 ± 0.000 |
| | GEM | 0.036 ± 0.006 | 0.070 ± 0.009 | 0.109 ± 0.011 | 0.135 ± 0.009 | 0.187 ± 0.008 | 0.241 ± 0.008 | 0.295 ± 0.008 |
| | RAP++ | 0.052 ± 0.001 | 0.088 ± 0.004 | 0.124 ± 0.008 | 0.146 ± 0.011 | 0.190 ± 0.013 | 0.234 ± 0.015 | 0.278 ± 0.015 |
| | PrivGSD | 0.036 ± 0.007 | 0.060 ± 0.011 | 0.084 ± 0.012 | 0.103 ± 0.011 | 0.141 ± 0.011 | 0.181 ± 0.011 | 0.221 ± 0.011 |
| | AIM | **0.028** ± **0.005** | **0.048** ± **0.009** | **0.072** ± **0.011** | **0.091** ± **0.008** | **0.128** ± **0.009** | **0.167** ± **0.009** | **0.206** ± **0.009** |
| | Tab-PE | 0.030 ± 0.004 | 0.051 ± 0.007 | 0.076 ± 0.009 | 0.095 ± 0.007 | 0.132 ± 0.008 | 0.170 ± 0.008 | 0.208 ± 0.009 |
| NN | *UB* | *0.024* ± *0.006* | *0.042* ± *0.009* | *0.065* ± *0.011* | *0.083* ± *0.012* | *0.119* ± *0.012* | *0.156* ± *0.012* | *0.192* ± *0.012* |
| | PrivSyn | 0.028 ± 0.004 | 0.065 ± 0.008 | 0.108 ± 0.010 | 0.149 ± 0.009 | 0.201 ± 0.011 | 0.256 ± 0.013 | 0.309 ± 0.014 |
| | PrivMRF | 0.030 ± 0.003 | 0.051 ± 0.006 | 0.076 ± 0.008 | 0.093 ± 0.009 | 0.131 ± 0.011 | 0.171 ± 0.013 | 0.211 ± 0.014 |
| | GEM | 0.037 ± 0.010 | 0.072 ± 0.012 | 0.115 ± 0.015 | 0.165 ± 0.029 | 0.219 ± 0.026 | 0.275 ± 0.024 | 0.330 ± 0.023 |
| | RAP++ | 0.050 ± 0.006 | 0.086 ± 0.011 | 0.119 ± 0.016 | 0.137 ± 0.016 | 0.179 ± 0.018 | 0.221 ± 0.020 | 0.263 ± 0.022 |
| | PrivGSD | 0.036 ± 0.003 | 0.058 ± 0.004 | 0.083 ± 0.004 | 0.108 ± 0.006 | 0.147 ± 0.004 | 0.187 ± 0.004 | 0.227 ± 0.005 |
| | AIM | **0.024** ± **0.002** | **0.042** ± **0.003** | **0.064** ± **0.003** | 0.094 ± 0.002 | 0.129 ± 0.002 | 0.165 ± 0.003 | 0.201 ± 0.003 |
| | Tab-PE | 0.028 ± 0.005 | 0.048 ± 0.008 | 0.071 ± 0.010 | **0.089** ± **0.010** | **0.125** ± **0.010** | **0.163** ± **0.010** | **0.199** ± **0.010** |
| RFF | *UB* | *0.020* ± *0.004* | *0.035* ± *0.005* | *0.056* ± *0.005* | *0.075* ± *0.004* | *0.113* ± *0.003* | *0.152* ± *0.003* | *0.191* ± *0.002* |
| | PrivSyn | 0.026 ± 0.004 | 0.058 ± 0.005 | 0.097 ± 0.006 | 0.120 ± 0.006 | 0.173 ± 0.005 | 0.227 ± 0.005 | 0.281 ± 0.005 |
| | PrivMRF | 0.023 ± 0.005 | 0.041 ± 0.008 | 0.064 ± 0.008 | 0.084 ± 0.005 | 0.122 ± 0.005 | 0.162 ± 0.005 | 0.202 ± 0.005 |
| | GEM | 0.036 ± 0.005 | 0.068 ± 0.008 | 0.105 ± 0.010 | 0.135 ± 0.020 | 0.187 ± 0.019 | 0.243 ± 0.019 | 0.298 ± 0.018 |
| | RAP++ | 0.050 ± 0.005 | 0.084 ± 0.008 | 0.116 ± 0.009 | 0.136 ± 0.007 | 0.178 ± 0.006 | 0.221 ± 0.005 | 0.263 ± 0.005 |
| | PrivGSD | 0.032 ± 0.002 | 0.053 ± 0.002 | 0.078 ± 0.002 | 0.098 ± 0.005 | 0.137 ± 0.004 | 0.179 ± 0.004 | 0.220 ± 0.003 |
| | AIM | 0.026 ± 0.002 | 0.045 ± 0.002 | 0.067 ± 0.002 | 0.088 ± 0.004 | 0.125 ± 0.004 | 0.165 ± 0.005 | 0.204 ± 0.005 |
| | Tab-PE | **0.022** ± **0.006** | **0.039** ± **0.009** | **0.061** ± **0.009** | **0.081** ± **0.008** | **0.120** ± **0.008** | **0.160** ± **0.008** | **0.198** ± **0.008** |

*Table 6.* The experiment is configured with $\epsilon = 1.0$. The degree hyperparameter of baselines varies from 2 to 5. The best and second-best results are highlighted in **bold** and underline, respectively.

### D.3. Real-world Datasets with High-Order Correlations

Table 7 presents the Wasserstein distances across low- to high-order ones. Consistently, Tab-PE outperforms the baselines on high-order Wasserstein distances (5-WD, 6-WD, and 7-WD). This indicates that Tab-PE is more effective in capturing high-order correlations.

| Dataset | Method | Fidelity (↓) | | | | | | |
|---|---|---|---|---|---|---|---|---|
| | | 1-WD | 2-WD | 3-WD | 4-WD | 5-WD | 6-WD | 7-WD |
| Artificial Characters | *UB* | *0.013* ±0.005 | *0.027* ±0.008 | *0.046* ±0.010 | *0.068* ±0.010 | *0.088* ±0.011 | *0.107* ±0.011 | *0.124* ±0.012 |
| | PrivSyn | 0.016 ±0.003 | 0.053 ±0.004 | 0.103 ±0.004 | 0.162 ±0.004 | 0.224 ±0.005 | 0.287 ±0.005 | 0.347 ±0.006 |
| | PrivMRF | 0.013 ±0.002 | 0.050 ±0.003 | 0.099 ±0.003 | 0.157 ±0.003 | 0.218 ±0.004 | 0.279 ±0.004 | 0.337 ±0.004 |
| | GEM | 0.091 ±0.007 | 0.152 ±0.010 | 0.211 ±0.012 | 0.273 ±0.013 | 0.337 ±0.014 | 0.402 ±0.015 | 0.465 ±0.015 |
| | RAP++ | 0.039 ±0.010 | 0.072 ±0.015 | 0.111 ±0.017 | 0.156 ±0.019 | 0.201 ±0.021 | 0.243 ±0.023 | 0.283 ±0.024 |
| | PrivGSD | 0.026 ±0.000 | 0.047 ±0.001 | **0.078** ±0.001 | **0.118** ±0.002 | 0.161 ±0.002 | 0.204 ±0.002 | 0.245 ±0.002 |
| | AIM | **0.011** ±0.000 | **0.040** ±0.001 | 0.082 ±0.001 | 0.134 ±0.001 | 0.191 ±0.003 | 0.247 ±0.002 | 0.301 ±0.002 |
| | Tab-PE | 0.029 ±0.006 | 0.056 ±0.009 | 0.087 ±0.010 | 0.123 ±0.011 | **0.158** ±0.011 | **0.191** ±0.012 | **0.220** ±0.013 |
| Person Activity | *UB* | *0.011* ±0.001 | *0.024* ±0.001 | *0.046* ±0.001 | *0.075* ±0.001 | *0.108* ±0.001 | *0.142* ±0.001 | *0.176* ±0.001 |
| | PrivSyn | 0.010 ±0.002 | 0.060 ±0.002 | 0.134 ±0.003 | 0.218 ±0.003 | 0.303 ±0.003 | 0.385 ±0.004 | 0.463 ±0.004 |
| | PrivMRF | **0.008** ±0.001 | 0.025 ±0.001 | 0.055 ±0.001 | 0.094 ±0.002 | 0.138 ±0.003 | 0.185 ±0.004 | 0.233 ±0.004 |
| | GEM | 0.065 ±0.003 | 0.114 ±0.001 | 0.164 ±0.002 | 0.218 ±0.003 | 0.275 ±0.005 | 0.333 ±0.006 | 0.392 ±0.007 |
| | RAP++ | 0.034 ±0.001 | 0.063 ±0.001 | 0.097 ±0.002 | 0.135 ±0.002 | 0.176 ±0.003 | 0.216 ±0.003 | 0.256 ±0.004 |
| | PrivGSD | 0.032 ±0.002 | 0.057 ±0.002 | 0.088 ±0.001 | 0.124 ±0.001 | 0.161 ±0.001 | 0.199 ±0.001 | 0.237 ±0.002 |
| | AIM | 0.009 ±0.001 | **0.023** ±0.001 | **0.049** ±0.001 | 0.085 ±0.002 | 0.125 ±0.002 | 0.168 ±0.002 | 0.213 ±0.002 |
| | Tab-PE | 0.012 ±0.003 | 0.026 ±0.003 | 0.050 ±0.003 | **0.082** ±0.002 | **0.116** ±0.002 | **0.150** ±0.002 | **0.183** ±0.001 |

*Table 7.* $\epsilon = 1.0$. Additional results on fidelity.

### D.4. Real-world Datasets with Low-Order Correlations

We further evaluate the methods on well-known real-world datasets with low-order correlations. Compared to Adult, Breast Cancer is a more challenging dataset with 30 features and only ∼500 samples. In this setting, we configure Tab-PE to run for 30 iterations generating 2K samples and 20 iterations generating 100 samples, respectively for the Adult and Breast Cancer datasets. The results are presented in Table 8. Consistent with the prior evaluations (Chen et al., 2025b), AIM offers the leading performance across most metrics. Tab-PE remains competitive on the downstream utilities with only 1% accuracy drop compared to AIM. For low-order correlations, the marginal-based methods are sufficient to capture the essential relationships between features and labels. This explains why AIM, RAP, GSD, and PrivMRF perform well on these datasets. Overall, these results indicate that while Tab-PE is primarily designed for high-order correlations, it remains competitive on datasets dominated by low-order correlations.

### D.5. Compute Efficiency

We present the running time and test accuracy of the baselines while varying the degree of marginal queries in Figure 15. Generally, increasing the degree of marginal queries does not significantly improve the accuracy. As the degree increases, the number of queries grows exponentially. For PrivMRF, the test performance peaks at the degree of 4 at 84% and remains stable at 83.5%. For GEM, the accuracy drops as the degree increases, due to the high noise to answer the large number of queries. The running time of GEM significantly increases at the degree of 5. For RAP++, the accuracy slightly increases from 84 to 84.45 at the degree of 4 then drops as the degree increases. The running time of RAP++ is not significantly affected by the degree of queries. For AIM, the accuracy remains approximately at 82% for all degrees that are less than 6. A degree that is too high ($\geq 6$) causes the method to collapse without producing any meaningful patterns. For GSD, the maximum degree is 3 due to the high compute resource requirement. In particular, at the degree of 4, GSD requires more than 200GB of GPU memory, which is not affordable for us. Theoretically, the running time of methods that rely on marginal queries grows exponentially as the number of queries increases. However, in practice, the running time of some methods is still affordable and do not change significantly. This is because some implementations limit the number of queries to a fixed number to make sure the noise is not too large to yield reliable query answers. As a result, they may not be able to fully model the high-order correlations. In summary, these results indicate the limitations of marginal-based methods in both efficiency and effectiveness in capturing high-order correlations.

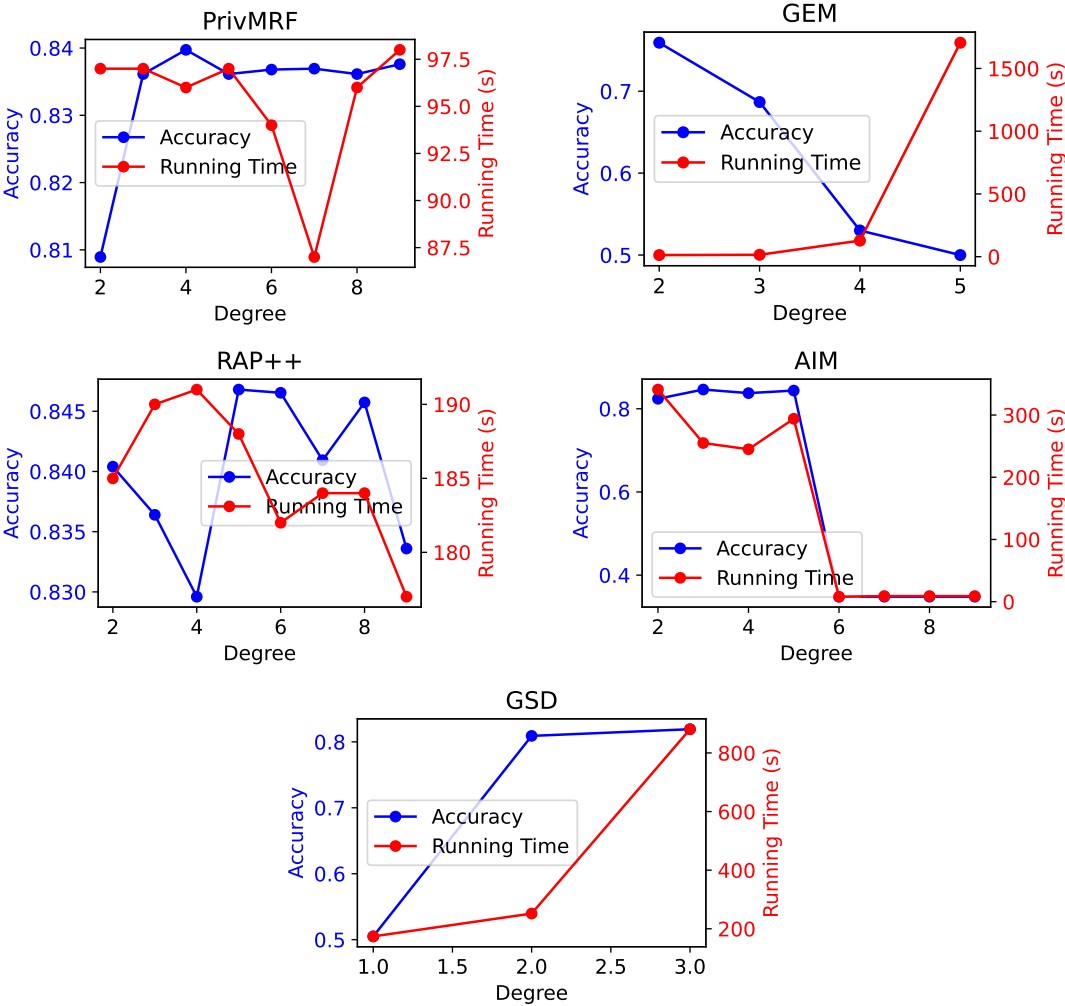

*Figure 15.* Running Time and Test Accuracy of the baselines while varying the degree of marginal queries. The trivial accuracy (random guessing) is 50%.

| Dataset | Method | ML Downstream (↑) | | Fidelity (↓) | | | Embedding (↑) | |
|---------|--------|----------|----------|------|------|------|-----------|--------|
| | | Accuracy | Macro F1 | 1-WD | 2-WD | 3-WD | Precision | Recall |
| Adult | *UB* | *84.41* ± *0.57* | *75.68* ± *1.54* | *0.014* ± *0.003* | *0.027* ± *0.006* | *0.041* ± *0.008* | *94.50* ± *0.14* | *94.09* ± *0.29* |
| | PrivSyn | 75.77 ± 0.00 | 43.11 ± 0.00 | **0.011** ± **0.002** | 0.034 ± 0.003 | 0.061 ± 0.003 | 45.34 ± 1.54 | 89.34 ± 0.17 |
| | PrivMRF | 83.15 ± 0.43 | **76.85** ± **0.64** | 0.017 ± 0.005 | **0.033** ± **0.009** | **0.052** ± **0.012** | 84.15 ± 0.25 | **93.74** ± **0.21** |
| | GEM | 79.17 ± 2.32 | 69.63 ± 1.48 | 0.037 ± 0.002 | 0.073 ± 0.005 | 0.109 ± 0.006 | 0.185 ± 0.004 | 76.48 ± 2.32 |
| | RAP++ | 80.87 ± 0.59 | 72.22 ± 1.25 | 0.023 ± 0.002 | 0.043 ± 0.003 | 0.066 ± 0.004 | 61.08 ± 4.24 | 80.64 ± 1.53 |
| | PrivGSD | 82.09 ± 0.11 | 75.90 ± 0.43 | 0.017 ± 0.001 | **0.033** ± **0.002** | **0.052** ± **0.003** | 74.45 ± 0.47 | 80.81 ± 0.41 |
| | AIM | **83.36** ± **0.41** | 76.10 ± 1.41 | 0.017 ± 0.001 | 0.034 ± 0.001 | 0.053 ± 0.001 | **87.06** ± **0.75** | 93.18 ± 0.21 |
| | Tab-PE | 82.22 ± 0.51 | 73.66 ± 0.87 | 0.049 ± 0.004 | 0.086 ± 0.007 | 0.118 ± 0.010 | 34.27 ± 1.57 | 77.25 ± 7.42 |
| Breast Cancer | *UB* | *97.68* ± *1.64* | *97.56* ± *1.73* | *0.078* ± *0.006* | *0.130* ± *0.011* | *0.175* ± *0.016* | *97.48* ± *0.73* | *95.64* ± *0.78* |
| | PrivSyn | 51.60 ± 8.03 | 38.99 ± 6.92 | 0.216 ± 0.010 | 0.353 ± 0.013 | 0.464 ± 0.014 | 60.39 ± 7.34 | 21.78 ± 7.84 |
| | PrivMRF | 60.41 ± 3.37 | 37.63 ± 1.33 | 0.180 ± 0.015 | 0.306 ± 0.022 | 0.410 ± 0.027 | 51.01 ± 11.29 | 16.83 ± 8.23 |
| | GEM | 50.74 ± 13.88 | 44.91 ± 12.27 | 0.181 ± 0.006 | 0.306 ± 0.007 | 0.409 ± 0.008 | **69.60** ± **11.83** | **28.64** ± **8.97** |
| | RAP++ | 84.81 ± 4.43 | 83.97 ± 4.45 | 0.227 ± 0.023 | 0.369 ± 0.032 | 0.483 ± 0.039 | 64.62 ± 3.48 | 9.63 ± 4.71 |
| | PrivGSD | 60.02 ± 3.13 | 37.49 ± 1.24 | 0.235 ± 0.013 | 0.379 ± 0.018 | 0.493 ± 0.021 | 60.30 ± 5.38 | 21.27 ± 1.91 |
| | AIM | **89.25** ± **4.75** | **87.82** ± **6.17** | 0.198 ± 0.009 | 0.332 ± 0.010 | 0.441 ± 0.011 | 68.17 ± 6.03 | 12.65 ± 4.86 |
| | Tab-PE | 88.48 ± 3.53 | 87.01 ± 4.74 | **0.162** ± **0.007** | **0.272** ± **0.010** | **0.366** ± **0.012** | 69.02 ± 6.14 | 15.24 ± 3.11 |

*Table 8.* Comparison on low-order real-world datasets under $\epsilon = 1$. The best and second-best results are highlighted in **bold** and underline, respectively. UB refers to the upper bound performance trained on real data.

### D.6. `VARIATION_API` Study

For consistency, this section presents the results on the Artificial Characters dataset with $\epsilon = 1.0$.

#### D.6.1. Decay Schedule Study

Figure 16 illustrates the linear and polynomial decay schedules for the mutation rate. Generally, the polynomial decay provides a higher mutation rate at the early stage for exploration while maintaining a smaller mutation rate at the later iterations for better refinement. This leads to better performance of the polynomial schedule over the linear decay, as shown in Figure 17.

**Polynomial schedule outperforms linear decay**. We study the impact of different probability decay schedules in the `VARIATION_API`. As shown in Fig. 17, App. D.6.1, the polynomial schedule consistently outperforms the linear decay, used in Lin et al. (2025), across all metrics. The polynomial schedule allows more aggressive exploration at the beginning and more focused refinement at the end, leading to better overall performance, illustrated in Fig. 16, App. D.6.1. We present additional performance analysis on different decay factors in App. D.6.1 (Fig. 22 and 23). Generally, a moderate initial mutation rate $\mu_{\text{init}}$ (0.5-0.7) and decay factor $\gamma$ (0.2 - 0.5) yield the best performance and consistently outperform the linear decay ($\gamma = 1.0$).

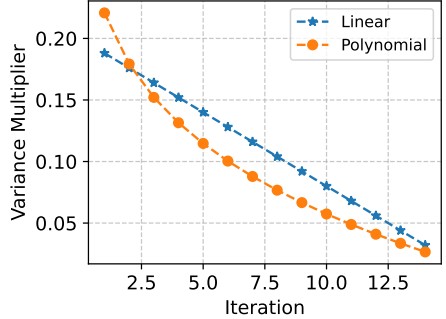

*Figure 16.* Visualization of different decay schedules.

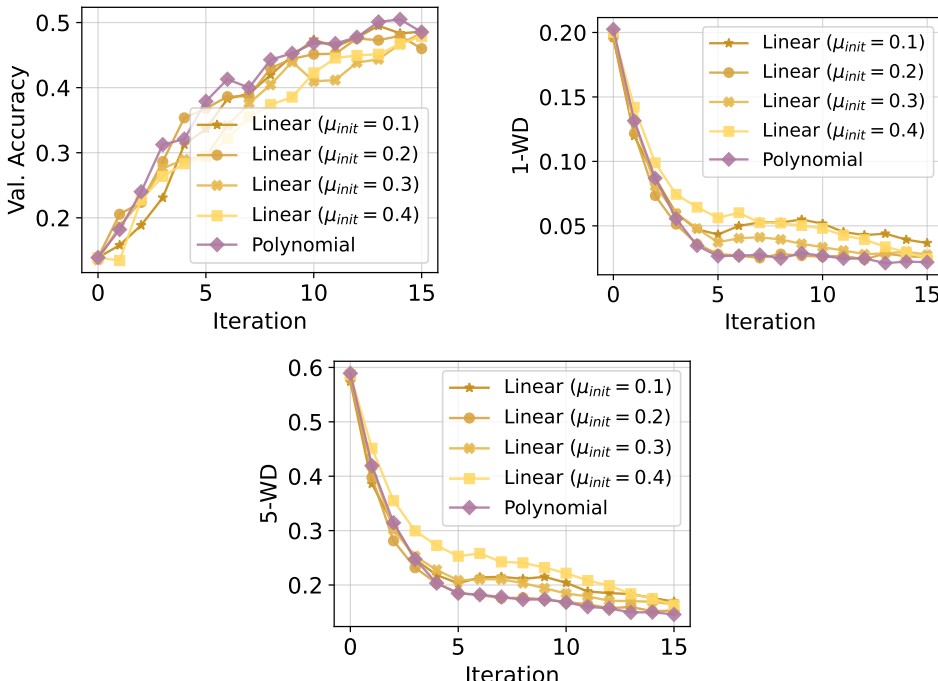

*Figure 17.* The performance of Tab-PE with different mutation rate decay schedules.

### D.6.2. VARIATION OPERATORS

We compare the proposed random walk strategy and a genetic algorithm-based design from an existing work – PrivGSD (Liu et al., 2023). It is worth noting there are significant differences between Private Evolution and PrivGSD. PrivGSD is a method that heavily relies on marginal queries. PrivGSD first defines a set of marginal queries and privately answers them. Then it uses a genetic algorithm to search for a synthetic dataset that minimizes the error compared to the noisy answers. Therefore, PrivGSD still inherits the limitations of marginal query-based approaches in capturing high-order correlations. In contrast, Tab-PE does not rely on marginal queries, our evolutionary process is directly guided by the private data at each iteration. In this comparison, we adapt the genetic algorithm-based design from PrivGSD to our `VARIATION_API` interface. More specifically, the original operators in PrivGSD work at dataset levels, while Tab-PE's `VARIATION_API` requires sample-level operators. To achieve this, we remove the random selection of samples from the dataset, and instead we apply the operators to all samples in the synthetic dataset. Additionally, PrivGSD performs mutation only one attribute at a time, which leads to very slow convergence (∼200K iterations). This is not affordable for Private Evolution, as the privacy budget is consumed at each iteration. Therefore, we modify the mutation operator to allow all attributes at once. The crossover operator is kept the same as in PrivGSD. Figure 6 presents the results. This confirms that the simple random-walk design in Tab-PE is effective and efficient.

### D.7. Extremely High-Dimensional Dataset (Flattened MNIST)

In this experiment, we rescale the original MNIST images from $28 \times 28$ to $14 \times 14$ and then flatten them into 196-dimensional vectors. We set the privacy budget $\epsilon$ to 1.0 and generate 300 synthetic images. We set the order of marginal queries to 2 for the baselines, as higher-order queries are not affordable for such a high-dimensional dataset. For Tab-PE, we run for 100 iterations with 30 sampling iterations and 20 variations. Such a large number of iterations and variations is necessary for this extremely high-dimensional dataset, as the search space is huge. For the classification accuracy, we employ a tiny CNN model with only 2 convolutional layers and 1 fully-connected layer. Fig. 18 provides some samples generated by the methods. It is expected that the marginal methods are not able to successfully reproduce the digit patterns, which requires extremely high-order understanding of pixels. In contrast, Tab-PE can capture the high-order correlations and generate samples that are visually similar to the real data. This demonstrates the effectiveness of Tab-PE in capturing high-order correlations even in extremely high-dimensional datasets.

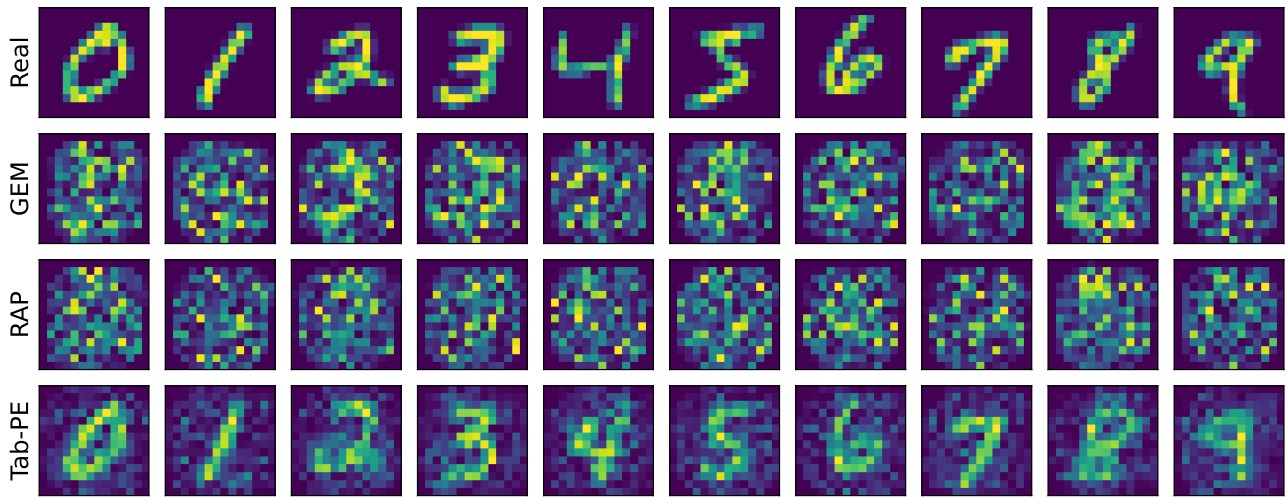

*Figure 18.* Samples generated by the methods for the flattened MNIST dataset with $\epsilon = 1.0$ . The first row corresponds to the real data. The remaining rows correspond to the synthetic data generated by the baselines and Tab-PE.

### D.8. Additional High-Order Real-World Datasets

We compare Tab-PE with AIM on several high-order real-world datasets, including Insurance[4], Monk[5], and Walking Activity[6]. Tab. 9 provides the results on the downstream utility. Tab-PE consistently outperforms AIM on all datasets, demonstrating the effectiveness of Tab-PE in capturing high-order correlations in real-world scenarios.

| Dataset | Method | Test Accuracy |
|---|---|---|
| Insurance | AIM | 89.86 |
| | Tab-PE | 93.24 |
| Monk | AIM | 61.54 |
| | Tab-PE | 64.84 |
| Walking Activity | AIM | 40.56 |
| | Tab-PE | 47.80 |

*Table 9.* Performance of Tab-PE and AIM on additional high-order real-world datasets.

### D.9. Hyperparameter Sensitivity Analysis

For consistency, this section presents the results on the Artificial Characters dataset with $\epsilon = 1.0$. For the hyperparameter sensitivity analysis, we vary one hyperparameter while keeping the others fixed as the default ones presented in the implementation if not specified. Due to expensive computational costs of calculating high-order Wasserstein distances, we use 1-WD only. The high-order Wasserstein distances' trends are usually consistent with the downstream utility, as shown in the main paper.

**Number of synthetic samples**    With fewer samples, the histogram counts are generally larger, which causes the noise to be less significant. However, too few samples may not be able to represent all high-dimensional correlations. If the number of samples is too large, the noise has more impact and can change the order of sample rankings. Figure 19 presents the performance of Tab-PE while varying the number of synthetic samples. At $\epsilon = 1.0$, 10% and 20% of the size of the private dataset achieve the best performance.

---

[4]https://www.kaggle.com/datasets/mosapabdelghany/medical-insurance-cost-dataset
[5]https://archive.ics.uci.edu/dataset/70/monk+s+problems
[6]https://archive.ics.uci.edu/dataset/286/user+identification+from+walking+activity

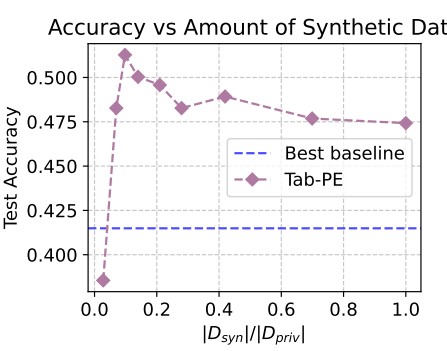
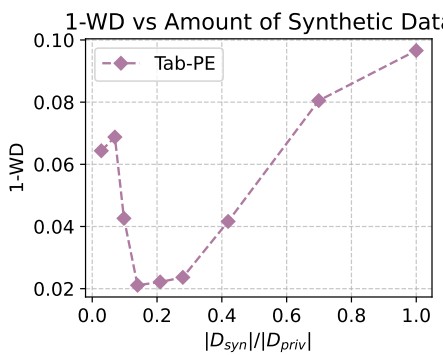

*Figure 19.* Tab-PE performance while varying the number of synthetic samples $\mathcal{D}_{\text{syn}}$.

**Number of iterations** A larger number of iterations $T$ allows more refinement of the synthetic data, but also leads to a larger noise scale $\sigma$. Figure 20 presents the performance of Tab-PE under different settings of the number of iterations $T$. Tab-PE needs around 15-20 iterations to achieve good performance.

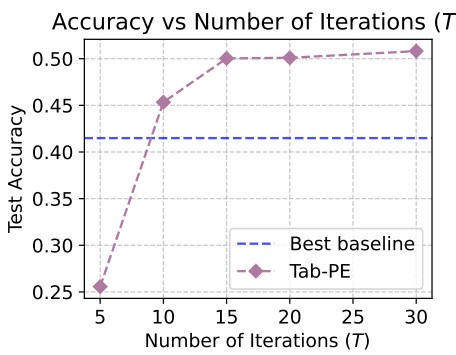
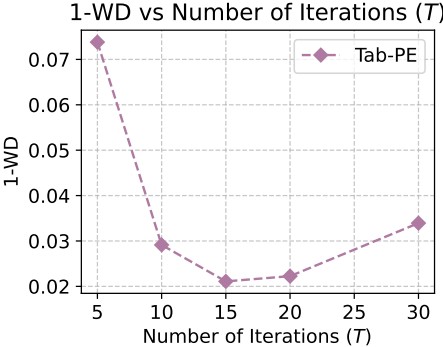

*Figure 20.* Tab-PE performance while varying the number of iterations $T$.

**Number of sampling iterations** The number of sampling iterations $T_{\text{sampling}}$ controls how many times we employ the sampling-with-replacement strategy. Figure 21 presents the performance of Tab-PE under different configurations of $T_{\text{sampling}}$. Generally, combining both sampling and ranking (i.e., $0 < T_{\text{sampling}} < T$) yields better performance than only ranking (i.e., $T_{\text{sampling}} = 0$) or only sampling (i.e., $T_{\text{sampling}} = T$). The best performance is achieved when using 20-40% of iterations for the sampling strategy.

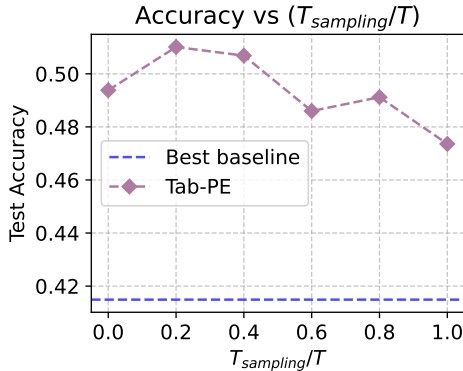
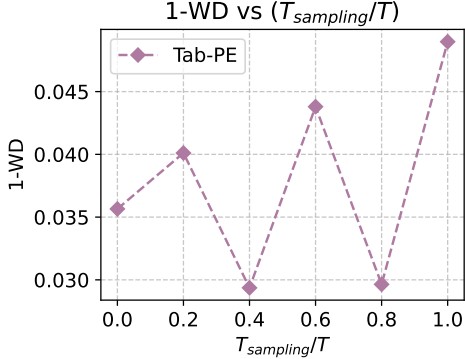

*Figure 21.* Tab-PE performance while varying the number of sampling iterations $T_{\text{sampling}}$.

**Mutation rate initial value** $\mu_{\text{init}}$    This parameter controls the noise level in the random walk strategy. A larger mutation rate enables more exploration, but also makes it harder for local refinement. Figure 22 presents the performance of Tab-PE with various values of $\mu_{\text{init}}$. A moderate value around 0.5-0.8 provides the best utility.

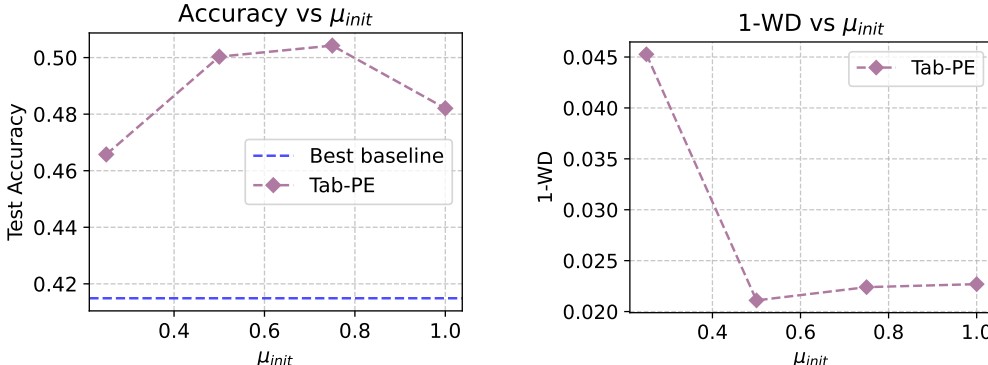

*Figure 22.* Tab-PE performance while varying the mutation initial rate $\mu_{init}$ in `VARIATION_API`.

**Decay factor** $\gamma$    This parameter controls how fast the mutation rate decays. A smaller $\gamma$ leads to a faster decay. A value at 1.0 is equal to a linear decay. Figure 23 presents the performance of Tab-PE with different settings of $\gamma$. A value around 0.5-0.75 achieves the best performance and outperforms the linear decay.

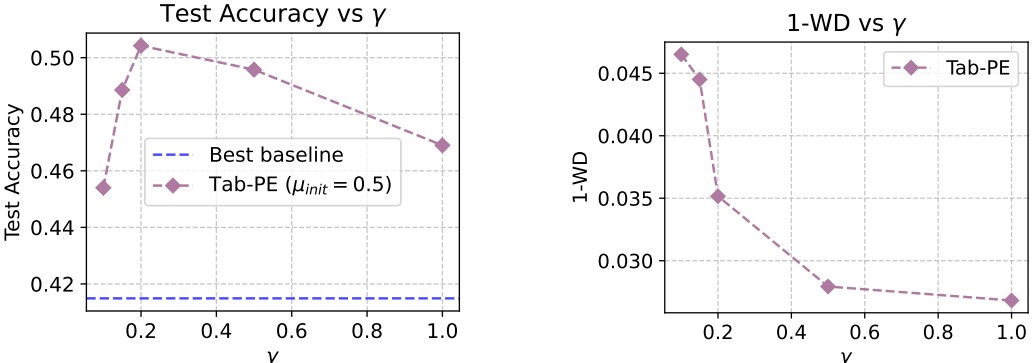

*Figure 23.* Tab-PE performance while varying $\gamma$ in the mutation rate schedule decay.

**Categorical-numerical weight** $\lambda$    This parameter controls the relative importance of categorical features and numerical features in the variation generation. A larger $\lambda$ means more focus on categorical features. A value of 0 means only numerical features are considered. Tab. 10 presents the performance of Tab-PE with different settings of $\lambda$. Without considering categorical features (i.e., $\lambda = 0$), the performance is significantly worse. The performance is fairly stable across different values of $\lambda$ that are larger than 0, which indicates the robustness of Tab-PE to this parameter. A moderate value around 0.333 achieves the best performance.

| $\lambda$ | 0.0 | 0.1 | 0.(333) | 1.0 |
|---|---|---|---|---|
| Val AUC | 0.510 | 0.630 | **0.631** | 0.630 |
| Val F1 | 0.266 | 0.350 | **0.354** | 0.348 |

*Table 10.* Performance of Tab-PE while varying the categorical weight $\lambda$ on the Artificial Characters dataset with $\epsilon = 1.0$.

**Hyperparameter search**    In Figure 24, we explore 144 settings of hyperparameters for the second stage (Tab-PE with ranking), the hyperparameters are chosen from the following sets: number of iterations (epochs) $\in \{15, 20, 30, 50\}$, num_samples $\in \{2k, 5k, 10k\}$, variation degree ($m = $ num_variations) $\in \{3, 5, 7\}$, and mutation rate initial value ($\mu_{init}$) $\in$

$\{0.15, 0.25, 0.35, 0.5\}$. The mutation rate in this experiment is set by a linear decay schedule. Note that from the figure, smaller values of all hyperparameters generally do better. This inspires us to employ the polynomial decay schedule for the mutation rate, which enables a larger mutation rate at early iterations and a smaller mutation rate at later iterations.

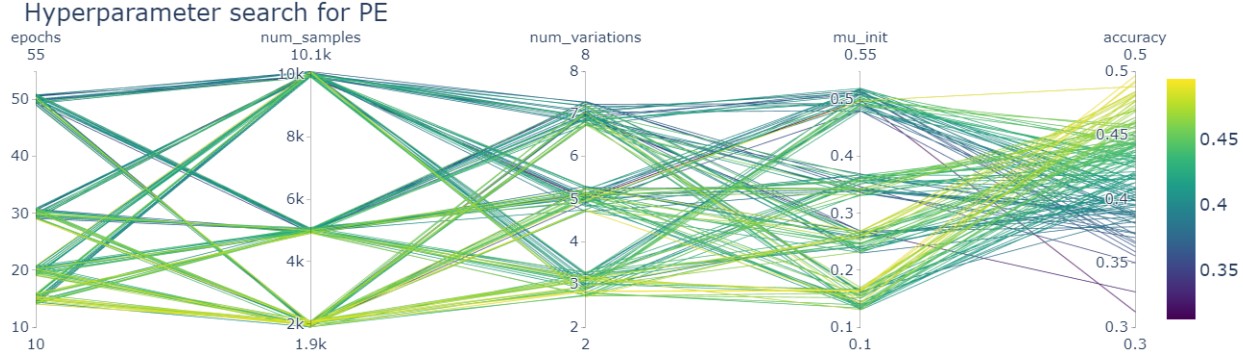

*Figure 24.* Hyperparameter search for Tab-PE on the Artificial Characters dataset for $\epsilon = 1.0$.

**Hyperparameter configuration across privacy settings** In Figure 25 for $\epsilon = 1.0$ (left-most column) we order all 144 hyperparameters according to their achieved accuracy. 0 corresponds to the best setting of hyperparameters, 1 corresponds to the second best setting, and so on. The lines are colored according to their performance on $\epsilon = 1.0$. The same line corresponds to the same setting of hyperparameters. We note that the same hyperparameters that are good for $\epsilon = 1.0$ are also good for $\epsilon = 3.0$ and $\epsilon = 10.0$.

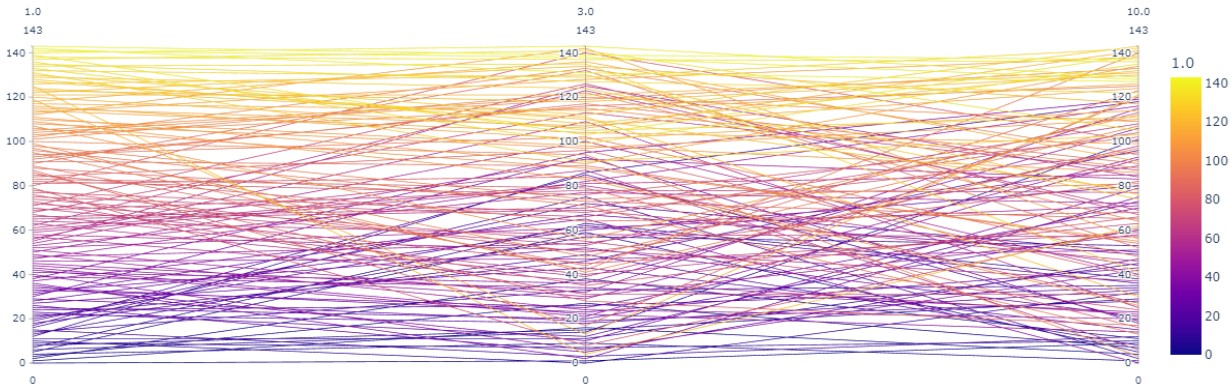

*Figure 25.* Ordering of the best hyperparameters for $\epsilon = 1.0$, $\epsilon = 3.0$ and $\epsilon = 10.0$.

### D.10. Oversampling Study

Following the simple recipe from PrivGSD (Liu et al., 2023), we conduct oversampling by randomly duplicating the samples. Figure 26 shows that the performance does not change significantly by oversampling.

### D.11. Noisy class distribution

We remove the assumption that the class distribution is public. Instead, we spend a bit of privacy budget to estimate the class distribution. To simplify, we spend $\epsilon_{\text{count}}$ out of $\epsilon$ for this estimation. Let $N_c$ be the count vector where $N_c[i]$ corresponds to the number of samples of class $i$. Since each sample is only counted once, the sensitivity of this counting process is 1. To achieve $(\epsilon_{\text{count}}, \delta)$-DP, we simply add noise, drawn from $\mathcal{N}(0, \sigma^2)$ to each count, where the noise multiplier is calculated by

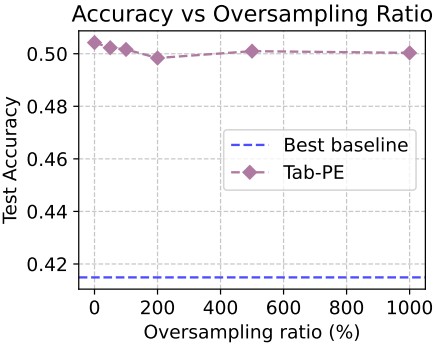

*Figure 26.* Tab-PE performance while enhancing by oversampling.

the analytic Gaussian Mechanism (Balle & Wang, 2018). In practice, our implementation uses the `diffprivlib` library to calculate this noise scale. We conduct an experiment spending 0.02 of the total privacy budget ($\epsilon = 1.0$) to privately estimate the class counts. Figure 27 presents the results of this experiment. Overall, the performance does not change much with the assumption that the class distribution is publicly available.

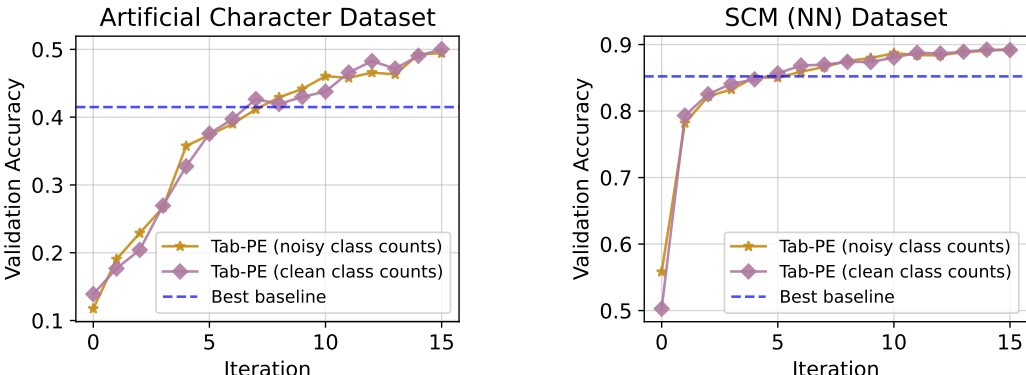

*Figure 27.* Tab-PE performance with noisy and clean class counts.

## E. Limitations & Future Work

Although the results are promising, there are still limitations. First, while Tab-PE consistently outperforms the baselines in capturing high-order correlations with better ML utilities, it underperforms on low-order fidelity, which primarily reflects low-order statistics. Second, the gap between Tab-PE and the upper bound (non-private) remains large. This gap is significantly larger than the current gaps of datasets with low-order correlations. Therefore, there is still room for further improvements. Third, Tab-PE currently implements a basic distance function on raw attribute scaled values without any embedding or attribute weighting. This can suffer in extreme cases where the data is sparse and most of the attributes are uninformative. While embedding in image and text domains can be achieved by pretrained foundation models, tabular data is very challenging because of strong distribution shifts across datasets. We leave this for future work. Additionally, we only explored simple designs of the Private Evolution APIs. More advanced APIs may further boost the performance.

