# OpenReview forum: "Differentially Private Synthetic Data via APIs 4: Tabular Data"
_ICML.cc/2026/Conference — ICML 2026 regular_

### Official Review · Reviewer_zfAP · 2026-03-08

**Soundness:** 2
**Presentation:** 3
**Significance:** 3
**Originality:** 2
**Overall Recommendation:** 4
**Confidence:** 3

**Summary:**

This paper proposed a differentially private tabular data synthesis algorithm called (Tab-PE) with Private Evolution (PE) framework. The proposed algorithm iteratively refines synthetic samples using simple perturbation operators and a differentially private nearest-neighbor histogram for scoring. The authors present better experimental results compared to the existing algorithm, e.g., AIM.

**Compliance With Llm Reviewing Policy:**

Affirmed.

**Final Justification:**

The rebuttal is reasonable for me. I would like to keep my positive score.

**Key Questions For Authors:**

1) Why does the proposed algorithm has relative inferior performance on the {1,2,3,4}-WD metrics?

2) How should one set the $\lambda$ parameter give a dataset? Is it possible to make the process DP as well?

3) Real-world experiments use at most 8 features. Is it possible to extend the experiments to datasets with more features?

Others can refer to the Weaknesses mentioned above.

**Limitations:**

Yes, there is an impact statement.

**Strengths And Weaknesses:**

Soundness:

[Strength 1] The technical execution is largely sound. The privacy analysis is correctly inherited from the original PE framework, with sensitivity arguments for the DP histogram.

[Strength 2] The experimental design is careful and includes the best existing DP tabular data synthesis mechanisms.

[Weakness 1] The hyperparameter $\lambda$ balancing categorical and numerical distance contributions is dataset-dependent and not selected via a privacy-preserving procedure, which is a minor but real gap given the paper's end-to-end DP framing.

[Weakness 2] While the fidelity results on {5, 6, 7}-WD look promising, the results on {1,2,3,4}-WD in Table 6 of the appendix show some gaps between the proposed algorithm and AIM (e.g., Artificial Characters, AIM achieves 1-WD of 0.011 while Tab-PE is at 0.029)

[Weakness 3] The core algorithm is a relatively direct adaptation of PE with simple tabular operators (Gaussian noise for numerical, resampling for categorical), offering limited algorithmic novelty beyond what PE already provides. Besides, while it seems the core of the algorithm is the nearest neighbor scoring mechanism, it is hard to tell how it can better implicitly capture high-order structure compared to the existing mechanisms.

Presentation:
The presentation of the paper is easy to follow in general.

Significance:
The proposed algorithm seems to have better performance on many metrics, but this becomes uncertain when it comes to low-dimensional WD fidelity metrics.

Originality:
The paper's main contribution is to adapt PE framework to tabular data.

---

> ### Author Rebuttal · Authors · 2026-03-31
>
> We kindly thank the reviewer for the constructive comments and suggestions. We provide clarifications and additional experimental results in the following.
>
> ## **Q3: Extremely high-order setting**
>
> We conduct an additional experiment on flattened MNIST, rescaled to 196 attributes. This setting can serve as a real-world stress test, as the number of attributes is large and the order of correlations is high, as digit shapes involve many pixels at the same time. We do not compare tabular methods with image-based methods (such as CNNs), which leverage the spatial structure (i.e., order of pixels) while tabular methods do not. At this scale, most workload-based methods are unfortunately not computationally feasible. Tab-PE significantly outperforms RAP and GEM either at loose or tight privacy budget. The result again demonstrates the limitations of query-based methods in high-order settings.
>
> | Method          | $\epsilon$ | Test Accuracy|
> |--------|---------|----------|
> |*Random Guess*| - | 10.00 |
> |*Private Set*| $\infty$ | 94.30 |
> |------|-----|-------|
> | GEM | 1.0 | 12.06 |
> | RAP | 1.0 | 9.82 |
> | Tab-PE | 1.0 | 54.05 |
> |------|-----|-------|
> | GEM | 50.0 | 22.09 |
> | RAP | 50.0 | 74.69 |
> | Tab-PE | 50.0 | 88.00 |
>
>
> ## **Q1, W2: Inferior performance on low-order metrics**
> The workload-based baselines are highly performant for {1,2,3}-WD as they explicitly optimize for the low-order metrics without considering high-order spectrum. For example, in the previous MNIST experiment, the workload-based methods can be very good at preserving the low-order correlations (e.g., single-pixel distributions) but fail to capture the high-order correlations (e.g., the joint distribution of multiple pixels that is necessary to form the digit shape). In contrast, Tab-PE can slightly underperform on the low-order metrics, but can capture the high-order correlations, which yields significantly better classification performance.
>
> Another example, consider a dataset with two binary features including 50% of "00" and 50% of "11". A method focusing on low-order marginals may generate samples with 50% of "01" and 50% of "10" and perfectly preserves 1-way marginals. Meanwhile, a method focusing on high-order correlations would generate 60% of "00" and 40% of "11", better capturing the joint distribution but slightly deviating from the 1-way marginals.
>
>
> ## **W1, Q2: Analysis on $\lambda$**
>
> We thank the reviewer for pointing this missing analysis. We have conducted additional experiments to analyze the impact of $\lambda$. The following table presents the performance of Tab-PE with different $\lambda$ values on the Person Activity dataset under $\epsilon=1.0$.
>
> | $\lambda$  | Val AUC  | Val F1  |
> |----------|--------|--------|
> | 0.0  |  0.510 |  0.266 |
> | 0.1   | 0.630  | 0.350  |
> | 0.3333   | 0.631  | 0.354  |
> | 1.0   | 0.630  | 0.348  |
>
> Without the categorical embedding component ($\lambda = 0$), the performance significantly drops. Tab-PE performs quite stably across different non-zero $\lambda$ values. In the experiments presented in the paper, we use a fixed $\lambda$ value of 0.3333 for all datasets. Even without any tuning on this hyperparameter, Tab-PE still outperforms the baselines.
>
>
> ## **Weakness 3: Algorithmic novelty and nearest neighbor matching**
>
> **Algorithmic novelty**: We appreciate this comment. We would like to highlight several points:
>
> The same core DP framework (nearest neighbor histogram with Gaussian noise) is shared across several published PE papers [1,2]. The novelty in each case lies in domain-specific adaptations and the new insights they enable, rather than the core algorithmic change.
>
> Additionally, Swanberg et al. (2025) show that PE with LLMs does not bring meaningful benefits for tabular data synthesis. Our work demonstrates that with careful design choices, PE can be highly effective. We believe this is a valuable insight of the PE paradigm and its applicability to different data types.
>
> We also hope the reviewer considers our contributions beyond the algorithm itself. We identify that commonly used benchmarks may overlook high-order correlations, which could partially explain why query-based methods have consistently appeared dominant in prior evaluations. We believe these insights are valuable for the community regardless of the specific algorithm.
>
> **Why Tab-PE is Effective for high-order structures**: The key difference lies in how information is accessed from the private data. While marginal-based methods decompose the joint order distribution into low-order projections, this decomposition can inherently lose high-order information. In contrast, Tab-PE with nearest neighbor matching considers *entire records*, naturally reflecting the full joint structure.
>
> [1] Differentially Private Synthetic Data via Foundation Model APIs 2: Text, Chulin Xie et al., ICML 2024
>
> [2] PrE-Text: Training Language Models on Private Federated Data in the Age of LLMs, Charlie Hou et al., ICML 2024

---

> > ### Author Rebuttal · Reviewer_zfAP · 2026-04-04
> >
> > Thanks for the response from the authors. I will keep my positive socore.

---

> > > ### Author Response · Authors · 2026-04-08
> > >
> > > Dear Reviewer,
> > >
> > > We appreciate your time and valuable feedback. We are glad that we have addressed your concerns.
> > >
> > > Best,
> > > Authors

---

### Official Review · Reviewer_qUno · 2026-03-10

**Soundness:** 3
**Presentation:** 4
**Significance:** 3
**Originality:** 3
**Overall Recommendation:** 5
**Confidence:** 4

**Summary:**

The paper presents a novel differentially private synthetic data generation algorithm for tabular data based on the Private Evolution framework.
Specifically, the authors design custom initialization, variation, and scoring functions for tabular data and run the PE algorithm.
In their testing they find that prior methods rarely capture high-order correlations in tabular data and their method outperforms prior work in simulated and real world datasets explicitly chosen with high-order correlations.

**Compliance With Llm Reviewing Policy:**

Affirmed.

**Final Justification:**

The authors addressed all my concerns during the rebuttal phase, therefore I improved my score to Accept

**Key Questions For Authors:**

1. How does the code handle the case where multiple synthetic samples are equally close to the private samples?
2. Are there any common real-world tabular datasets where the authors' PE method is better than prior work?
3. Why do marginal methods fail even when the total number of queries possible is small?

**Limitations:**

yes

**Strengths And Weaknesses:**

This paper was well-written and I enjoyed reading it.
The methodology and specifically the idea to use simple operators over fancy foundational models for tabular data is interesting.
The results also certainly look promising.
Nevertheless, I have a couple of comments I hope the authors will take into consideration.

1. There is a slight subtlety in the privacy analysis of DP_NN_HISTOGRAM. The argmin may not necessarily be unique, even when the synthetic records themselves are unique, since a private sample can be equidistant from two or more synthetic samples. I am not sure if this is a case explicitly handled in the code but this has to be clarified as the sensitivity of the algorithm depends on this.

2. The datasets used in the paper are either simulated or very uncommon. I understand the motivation behind the authors wanting to model high-order correlations but the question they have not sufficiently addressed in the work is whether or how much real-world tabular datasets actually display high-order correlations. In the paper, they present two real-world datasets, Artificial Characters and Person Activity, but the origin of these datasets is not really explained, thus making me wonder about their actual "real-world-ed-ness" (the name Artificial Characters does not help to be honest). I have two suggestions for the authors here, either (1) raise this as a limitation of the work, that the translation of these results into actual real-world datasets is currently unclear and left for future work or (2) test the algorithm against more commonly used real world datasets, I notice other datasets such as the FIRE dataset used in the NIST competition or the NYC taxi data used by Swanberg et al. 2025 might be good candidates since they are not demographic datasets.

3. I am slightly concerned about the results of the stress test. 3/4/5/6/7 binary features should not be too difficult for AIM to handle (they should total up to only 8/16/32/64/128 queries). So I am not sure why the marginal methods are failing so badly here, unless they have been artificially restricted to only analyzing 1/2-way marginals in this case.

---

> ### Author Rebuttal · Authors · 2026-03-31
>
> We appreciate the reviewer for the constructive comments and suggestions. We are very happy that the reviewer enjoyed reading the paper.
>
> ## **Q1: Argmin in DP_NN_HISTOGRAM**
> We thank the reviewer for pointing out this great detail. It is feasible that the argmin may not be unique. In our implementation, only a single nearest neighbor is selected (the first occurrence as we used numpy.argmin). This ensures the sensitivity of the algorithm is exactly 1. We will clarify this detail in the revised version.
>
> ## **Q2: Additional real-world datasets**
>
> We appreciate the reviewer's insightful comment and suggestions. We acknowledge that the majority of tabular datasets are dominated by low-order correlations, which are not the focus of our work. We will add a discussion on this realness aspect in the limitation section and encourage future work to investigate this further.
>
> It is worth noting that we have some results on well-known low-order datasets in Appendix D.4 (Table 7), where Tab-PE performs competitively. Additionally, the URL links to the two datasets (Artificial Characters and Person Activity) are provided in Appendix C.1.1.
>
> The following table presents experimental results on the Insurance dataset [1], which includes 7 attributes. Tab-PE outperforms leading baselines (AIM and GSD) on this dataset.
>
> | Method          | $\epsilon$ | Test Accuracy|
> |------|------|----|
> |*Random Guess*| - | 50.12 |
> |*Private Set*| - | 96.87 |
> | AIM | 1.0 | 89.86 |
> | GSD | 1.0 | 86.96 |
> | Tab-PE | 1.0 | 93.24 |
> | GEM | 3.0 | 93.72 |
> | RAP | 3.0 | 94.69 |
> | Tab-PE | 3.0 | 96.38 |
>
> The NYC taxi dataset is significantly large (2GB) and does not include a clear classification task. We were unfortunately unable to find the FIRE dataset.
>
> We conduct an additional experiment on flattened MNIST, rescaled to 196 attributes. This setting can serve as a real-world stress test, as the number of attributes is large and the order of correlations is high, as digit shapes involve many pixels at the same time. We do not compare tabular methods with image-based methods (such as CNNs), which leverage the spatial structure (i.e., order of pixels) while tabular methods do not.  At this scale, most workload-based methods are unfortunately not computationally feasible. Tab-PE significantly outperforms RAP and GEM either at loose or tight privacy budget. The result again demonstrates the limitations of query-based methods in high-order settings.
>
> | Method          | $\epsilon$ | Test Acc|
> |-----|------|------|
> |*Random Guess*| - | 10.00 |
> |*Private Set*| - | 94.30 |
> | GEM | 1.0 | 12.06 |
> | RAP | 1.0 | 9.82 |
> | Tab-PE | 1.0 | 54.05 |
> | GEM | 50.0 | 22.09 |
> | RAP | 50.0 | 74.69 |
> | Tab-PE | 50.0 | 88.00 |
>
> We also do experiments on several other high-order datasets from UCI, Tab-PE can enhance the performance compared to the baselines.
>
> | Method          | Dataset | Test Acc|
> |-------|------|-----|
> | AIM | monk | 61.54 |
> | Tab-PE | monk | 64.84 |
> | AIM | walking-activity | 40.56 |
> | Tab-PE | walking-activity | 47.80 |
>
>
> We thank the reviewer for the kind suggestions. We will add a discussion on this realness aspect in the limitation section and encourage future work to investigate this further.
>
> [1] Insurance: https://www.kaggle.com/datasets/mosapabdelghany/medical-insurance-cost-dataset
>
> [2] Monk: https://archive.ics.uci.edu/dataset/70
>
> [3] Walking Activity: https://archive.ics.uci.edu/dataset/286
>
> ## **Q3: Stress-test clarification**
>
> We thank the reviewer for this important clarification question. We would like to clarify that the query order is not restricted but set to the *ideal* order (K = num_attributes + 1), enabling a chance to capture full-order joint distributions.  It is worth noting that in practice, the ideal order is unknown. First, the marginal methods generally consider not only K-way marginals but also all lower-order marginals (i.e., K=1, 2, .., K). AIM iteratively selects the most beneficial marginal among the query pool based on noisy measurements. We observe that even when AIM is configured with the ideal order, it still fails to pick the right marginals to query, leading to poor performance.
>
> Additionally, marginal methods require to discretize the continuous features. We do not configure the baselines with the ideal discretization, as this information is not available in practice. A popular way is to use a uniform discretization with a fine granularity (e.g., N = 16 bins (PrivBayes), 32 bins (AIM)), but it can lead to a rapid explosion in the number of queries quickly ($N^K$ instead of $2^K$). Instead, we employ PrivTree, which adaptively partitions feature space and has been shown to be the most effective method [1]. Nevertheless, there is no guarantee that PrivTree can find the optimal discretization, resulting to more queries than the optimal $2^K$. In contrast, Tab-PE does not require discretization and directly works with continuous features, which can be an advantage in this setting.
>
> [1] arXiv:2504.06923

---

> > ### Author Rebuttal · Reviewer_qUno · 2026-04-01
> >
> > I thank the authors for their response to all of my concerns. I will improve my score accordingly.

---

> > > ### Author Response · Authors · 2026-04-08
> > >
> > > Dear Reviewer,
> > >
> > > We are glad that we have resolved your concerns. Thank you again for your time and insightful feedback!!
> > >
> > > Best,
> > > Authors

---

### Official Review · Reviewer_oTFa · 2026-03-13

**Soundness:** 3
**Presentation:** 3
**Significance:** 3
**Originality:** 2
**Overall Recommendation:** 4
**Confidence:** 3

**Summary:**

This paper studies differentially private synthetic tabular data generation in settings where high-order correlations matter, and proposes Tab-PE, a tabular adaptation of Private Evolution with lightweight heuristic operators and DP nearest-neighbor scoring. The main empirical claim is that, on simulated and selected real datasets with stronger high-order structure, Tab-PE achieves better downstream utility than prior baselines while being substantially more efficient.

**Compliance With Llm Reviewing Policy:**

Affirmed.

**Final Justification:**

I believe that all my concerns have been addressed. I would be happy to keep my positive score.

**Key Questions For Authors:**

Please see weaknesses.

**Limitations:**

yes

**Strengths And Weaknesses:**

## Strengths

- The paper makes a plausible case that many commonly used DP tabular benchmarks are dominated by low-order dependencies, which can hide the limitations of marginal-query methods on higher-order structure.
- The paper includes XOR stress tests, SCM simulations with three priors, two real datasets, privacy-budget sweeps, runtime and scalability analysis, and several appendix ablations. This gives the paper a reasonably complete experimental story.
- The baselines are tuned over query degree rather than fixed to an artificially weak setting, and the appendix discusses additional analyses on hyperparameters, variation design, and the class-distribution assumption.
- The efficiency result is a genuine practical advantage. The method avoids model training, runs on CPUs, and is reported to outperform AIM on the targeted settings while being much faster.

## Weaknesses

- The paper ultimately focuses on only two real datasets selected as high-order, so the broader claims about practical tabular settings remain somewhat stronger than the evidence warrants.
- In Section 2, the paper explicitly discusses PrivPGD and JAM as closely related iterative or query-based methods, and it also highlights Swanberg et al. as the most directly relevant prior attempt to apply Private Evolution to tabular data. However, none of these methods appear in the actual baseline suite in Section 5.1, where the comparisons are limited to PrivSyn, PrivMRF, GEM, RAP++, PrivGSD, and AIM. In particular, omitting Swanberg et al. makes it harder to judge whether the gains come from the PE paradigm itself or from the specific tabular design choices introduced here.
- The appendix states that hyperparameter tuning uses a validation set and that its privacy cost is generally not accounted for. That assumption is common, but it matters here because the paper emphasizes performance in strict privacy regimes.
- The method is a sensible and practically useful tabular adaptation of PE, but the core algorithmic change relative to the original PE framework is limited.

---

> ### Author Rebuttal · Authors · 2026-03-31
>
> We thank the reviewer for the constructive comments and suggestions.
>
> ## W1: Additional real-world datasets
>
> The following table presents experimental results on the Insurance dataset [1]. Tab-PE outperforms leading baselines on this dataset.
>
> | Method          | $\epsilon$ | Test Accuracy|
> |----------------|---------------|---------------|
> |*Random Guess*| - | 50.12 |
> |*Private Set*| - | 96.87 |
> | AIM | 1.0 | 89.86 |
> | GSD | 1.0 | 86.96 |
> | Tab-PE | 1.0 | 93.24 |
> | GEM | 3.0 | 93.72 |
> | RAP | 3.0 | 94.69 |
> | Tab-PE | 3.0 | 96.38 |
>
> We conduct an additional experiment on flattened MNIST, rescaled to 196 attributes. This setting can serve as a real-world stress test, as the number of attributes is large and the order of correlations is high, as digit shapes involve many pixels at the same time. We do not compare tabular methods with image-based methods (such as CNNs), which leverage the spatial structure (i.e., order of pixels) while tabular methods do not. At this scale, most workload-based methods are unfortunately not computationally feasible. Tab-PE significantly outperforms RAP and GEM either at loose or tight privacy budget. The result again demonstrates the limitations of query-based methods in high-order settings.
>
> | Method          | $\epsilon$ | Test Accuracy|
> |----------------|---------------|---------------|
> |*Random Guess*| - | 10.00 |
> |*Private Set*| - | 94.30 |
> | GEM | 1.0 | 12.06 |
> | RAP | 1.0 | 9.82 |
> | Tab-PE | 1.0 | 54.05 |
> | GEM | 50.0 | 22.09 |
> | RAP | 50.0 | 74.69 |
> | Tab-PE | 50.0 | 88.00 |
>
> We also do experiments on serveral other high-order datasets from UCI, Tab-PE can enhance the performance compared AIM.
>
> | Method  ($\epsilon = 1$)         | Dataset | Test Accuracy|
> |----------------|---------------|---------------|
> | AIM | [2]| 61.54 |
> | Tab-PE | [2]| 64.84 |
> | AIM | [3] | 40.56 |
> | Tab-PE | [3] | 47.80 |
>
>
> [1] Insurance: https://www.kaggle.com/datasets/mosapabdelghany/medical-insurance-cost-dataset
>
> [2] Monk: https://archive.ics.uci.edu/dataset/70/monk+s+problems
>
> [3] Walking Activity: https://archive.ics.uci.edu/dataset/286/user+identification+from+walking+activity
>
> ## W2: Baseline clarification
> We thank the reviewer for highlighting these works. We clarify that the mentioned methods (PrivPGD, JAM, and Swanberg et al.) were excluded due to incompatibility with our setting or inferior performance compared to our existing baselines. More specifically, JAM assumes some available public data, which is not permitted in our setting. Swanberg et al. requires expensive LLMs (e.g., Gemini). The main takeaway from Swanberg et al. is that LLMs do not bring meaningful benefits for tabular data synthesis. The paper provides a valuable and insightful exploration, but it does not improve upon the SOTA methods. Therefore, we did not include it in our baselines. Moreover, the most recent benchmark (SIGMOD 2026) [1] confirms that AIM and PrivGSD are still the leading methods, both of which are included in our baselines.
>
> PrivPGD does not outperform our existing best baselines.
> | Method          | Dataset | Test Accuracy|
> |----------------|---------------|---------------|
> | PrivPGD | Person Activity | 50.86 |
> | AIM | - | 59.56 |
> | PrivGSD | - | 54.25 |
> | TabPE | - | 62.88 |
> | PrivPGD | Artificial Characters | 26.94 |
> | AIM | - | 22.76 |
> | PrivGSD | - | 41.49 |
> | TabPE | - | 48.08 |
>
> [1] arXiv:2504.14061
>
> ## W3: Hyperparameter tuning
> We thank the reviewer for the comment. While we do not account for the privacy cost of hyperparameter tuning, all the baselines and Tab-PE are tuned with the same procedure. This ensures a fair comparison across all methods. This is also a common practice in the DP data synthesis literature [1-4].
>
> [1-4] arXiv:2106.07153, 1802.06739, 2306.03257, 2103.06641
>
> ## W4: Algorithmic novelty
>
> We appreciate this comment. We would like to highlight several points:
>
> The same core DP framework (nearest neighbor histogram with Gaussian noise) is shared across several published PE papers [1,2]. The novelty in each case lies in domain-specific adaptations and the new insights they enable, rather than the core algorithmic change.
>
> Additionally, Swanberg et al. show that PE with LLMs does not bring meaningful benefits for tabular data synthesis. Our work demonstrates with careful design choices, PE can be highly effective. We believe this is a valuable insight of the PE paradigm and its applicability to different data types.
>
> We also hope the reviewer considers our contributions beyond the algorithm itself. We identify that commonly used benchmarks may overlook high-order correlations, which could partially explain why query-based methods have consistently appeared dominant in prior evaluations. We believe these insights are valuable for the community regardless of the specific algorithm.
>
> [1] Differentially Private Synthetic Data via Foundation Model APIs 2: Text, ICML 2024
>
> [2] PrE-Text: Training Language Models on Private Federated Data in the Age of LLMs, ICML 2024

---

> > ### Author Rebuttal · Reviewer_oTFa · 2026-04-04
> >
> > Thank you for your response. I believe that all my concerns have been addressed. I would be happy to keep my positive score.

---

> > > ### Author Response · Authors · 2026-04-08
> > >
> > > Dear Reviewer,
> > >
> > > Thank you a lot for your time and constructive feedback. We are glad that your concerns have been addressed.
> > >
> > > Best,
> > > Authors

---

### Official Review · Reviewer_a6p1 · 2026-03-13

**Soundness:** 3
**Presentation:** 3
**Significance:** 3
**Originality:** 3
**Overall Recommendation:** 4
**Confidence:** 4

**Summary:**

This paper revisits differentially private (DP) tabular data synthesis with a specific focus on preserving high-order correlations. It argues that existing benchmarks are often dominated by low-order dependencies, which can lead to an overestimation of marginal/PGM-based methods. To address this, the paper proposes **Tab-PE**, a tabular adaptation of the Private Evolution (PE) framework. Tab-PE iteratively (1) generates candidate variations via lightweight tabular operators, (2) privately scores candidates using a DP nearest-neighbor histogram (where private records vote for the nearest candidate and Gaussian noise is added), and (3) selects the next generation via a two-stage strategy (sampling vs. top-$k$). Notably, Tab-PE does not rely on foundation models and operates efficiently on CPU. Experiments across XOR stress tests, SCM simulations, and real-world datasets demonstrate improved downstream utility and faster runtimes compared to baselines like AIM, RAP++, and PrivGSD.

**Compliance With Llm Reviewing Policy:**

Affirmed.

**Key Questions For Authors:**

1. **DP Accounting Specifics:** Each iteration releases a noisy histogram vector of length $|P|$. What specific accountant (e.g., RDP, Gaussian Accountant) is used? How exactly do $\sigma$ and $T$ map to the final $(\epsilon, \delta)$? If class-wise synthesis is applied, do you utilize parallel composition?
2. **Ablation of Selection Strategy:** Can you provide a comparison of the "two-stage selection" against "always-sampling" and "always-top-$k$" baselines? How sensitive are the results to the switch point $T_{\text{sampling}}$?
3. **Distance Robustness:** How do $\lambda$, numerical normalization, and missing-value handling affect the NN search? Is the performance stable across different feature scalings?
4. **Direct High-Order Validation:** In the XOR or SCM experiments, can you report a direct metric for interaction recovery (e.g., the accuracy of capturing the XOR relationship itself) rather than relying solely on downstream classifier performance?
5. **Replication Risk:** Can you include an empirical privacy-risk metric (e.g., Distance to Closest Record) to demonstrate that the improved utility does not stem from excessive replication of private samples?

**Limitations:**

Not fully. The paper motivates DP, but the “Limitations / Potential negative impact” discussion could be strengthened. I suggest adding: (1) a clearer statement that Tab-PE’s effectiveness depends on the choice of distance metric and may degrade under high-dimensional distance concentration, high-cardinality categoricals, missing-data patterns, or strong distributional shift; (2) an explicit discussion of potential memorization/replication risk due to sample-level nearest-neighbor matching (even under DP), ideally with at least one empirical risk check (e.g., nearest-neighbor distance / near-duplicate rate vs (\varepsilon), or membership/attribute inference); (3) a note on misuse risks (e.g., synthetic data for surveillance/targeting) and appropriate safeguards/usage guidance; and (4) limitations of evaluation (e.g., downstream metrics not fully capturing all high-order dependencies, and dependence on representation-space metrics).

**Strengths And Weaknesses:**

**Soundness.**
The proposed approach is technically reasonable and builds on a well-defined differentially private mechanism. The DP nearest-neighbor histogram scoring procedure provides a principled way to guide the evolutionary search while maintaining privacy guarantees. The experimental evaluation is extensive and includes both synthetic datasets specifically designed to exhibit strong high-order correlations (e.g., XOR-type structures) and multiple real-world datasets, with comparisons against several strong baselines. However, some components of the method (such as the distance metric for mixed-type features, mutation operators, and the two-stage selection strategy) appear largely heuristic, and the paper provides limited ablation analysis to isolate the contribution of these design choices.

**Presentation.**
Overall, the paper is clearly written and well structured. The motivation for focusing on high-order correlations in differentially private tabular synthesis is well explained, and the algorithmic pipeline is reasonably easy to follow. The related work discussion is also fairly comprehensive. That said, some implementation details—such as feature normalization in the distance metric and certain hyperparameter settings—are not fully described in the main text, which may make reproduction somewhat harder without consulting the supplementary material.

**Significance.**
Capturing high-order correlations in differentially private tabular data synthesis is an important and well-recognized challenge. The proposed approach offers a potentially scalable alternative to marginal-based methods that suffer from combinatorial explosion when modeling high-order dependencies. If validated more broadly, this idea could be useful for high-dimensional tabular datasets where traditional DP synthesis methods struggle.

**Originality.**
The paper proposes a novel combination of private evolutionary search with a DP nearest-neighbor scoring mechanism tailored to tabular data. While the individual components (evolutionary search, nearest-neighbor scoring, DP noise addition) are not entirely new, their integration into a practical framework for DP tabular synthesis is interesting. In particular, adapting the Private Evolution paradigm to work without relying on large pretrained models represents a creative and practically motivated contribution.

**Strengths:**

* **Timely Framing:** The focus on high-order correlations addresses a clear limitation in standard DP tabular benchmarks.
* **Practicality:** The method's efficiency (CPU-based, no foundation model) and scalability are significant advantages over many deep learning alternatives.
* **Empirical Breadth:** Robust evidence across multiple metrics (utility, fidelity, efficiency) and dataset regimes.

**Weaknesses:**

* **Heuristic Dependencies:** Key components (two-stage selection, distance design) lack systematic ablation, making it difficult to pinpoint the exact source of gains.
* **DP Accounting Presentation:** The presentation of the per-iteration release and class-wise synthesis accounting is currently fragmented between the main text and the appendix.
* **Robustness & Risk:** As the method relies on sample-level NN matching, the risk of "memorization" or nearest-neighbor leakage (even under DP) warrants empirical evaluation via replication metrics.

---

> ### Author Rebuttal · Authors · 2026-03-31
>
> We thank the reviewer for the constructive questions.
> ## **W1, Q2, Q3: Missing Analysis**
> We would like to clarify that the ablation study of two-stage selection is already presented in Appendix D.6 (Figure 14).
>
> Regarding the distance metric, we vary $\lambda$ on the Artificial Characters dataset under $\epsilon=1.0$. Tab-PE performs quite stably across different non-zero $\lambda$ values.
>
>
> | $\lambda$ | Val AUC | Val F1 |
> | --------- | ------- | ------ |
> | 0.0       | 0.510   | 0.266  |
> | 0.1       | 0.630   | 0.350  |
> | 0.3333    | 0.631   | 0.354  |
> | 1.0       | 0.630   | 0.348  |
>
>
> ## **W2, Q3: DP Accounting**
> Our DP analysis is presented in Appendix B.1 and reused from the original PE framework. Due to space constraints, we defer the full DP analysis to the Appendix and the original paper.
>
> More specifically, we reuse the implementation from DPSDA [1], which uses Analytic Gaussian Mechanism [2]. The class-wise synthesis is applied with parallel composition, which is also reflected in the implementation.
>
> [1] https://github.com/microsoft/DPSDA
>
> [2] https://arxiv.org/abs/1805.06530
>
> ## **Q3: Direct High-Order Validation**
>
> The below table presents the mutual information values of different combinations of the XOR dataset with 2 attributes $X_1$ and $X_2$. The mutual information $I(Y;X_1;X_2)$ is significantly larger than the ones of lower-order combinations. This indicates a 3-way correlation between $X_1, X_2$ and $Y$.
>
>
> | Combination    | Value  |
> | -------------- | ------ |
> | $I(X_1; X_2)$  | 0.0048 |
> | $I(Y;X_1)$     | 0.0004 |
> | $I(Y;X_2)$     | 0.0003 |
> | $I(Y;X_1;X_2)$ | 0.8970 |
>
>
> ## **W3, Q5: Replication Risk**
>
> We thank the reviewer for raising this concern. We will add a note in our guideline to make sure users are aware of the potential risks due to duplication. Here, we analyze the distance from each synthetic sample to its nearest neighbor in the private set. **Tab-PE does not show higher risks of replication compared to AIM**.
>
> | Class | AIM (min) | Tab-PE (min) |
> | ----- | --------- | ------------ |
> | 0     | 0.02      | 0.03         |
> | 1     | 0.03      | 0.04         |
> | 2     | 0.08      | 0.04         |
> | 3     | 0.01      | 0.03         |
> | 4     | 0.02      | 0.04         |
> | 5     | 0.06      | 0.03         |
> | 6     | 0.02      | 0.03         |
> | 7     | 0.03      | 0.02         |
> | 8     | 0.03      | 0.05         |
> | 9     | 0.02      | 0.04         |

---

### Decision · Program_Chairs · 2026-04-30

**Decision:**

Accept (regular)

**Comment:**

Reviewers agreed that the paper considers a reasonable problem and clearly presents a meaningful technical contribution to solve it. The authors should make sure to incorporate the discussion feedback, particularly the experiments on additional datasets and different values of $\lambda$.